# High-dimensional Mean-Field Games by Particle-based Flow-Matching

**Jiajia Yu[1], Junghwan Lee[2], Yao Xie[2], Xiuyuan Cheng[1]**
[1] Duke University, [2] Georgia Institute of Technology.

## Abstract

Mean-field games (MFGs) study the Nash equilibrium of systems with a continuum of interacting agents, which can be formulated as the fixed-point of optimal control problems. They provide a unified framework for a variety of problems, including both potential and non-potential games, with applications in areas such as generative modeling. Despite their broad applicability, solving high-dimensional MFGs remains a significant challenge due to fundamental computational and analytical obstacles. In this work, we propose a particle-based deep Flow Matching (FM) method to tackle high-dimensional MFG computation. In each iteration of our proximal fixed-point scheme, particles are updated using first-order information, and a flow neural network is trained to match the velocity of the sample trajectories. Theoretically, in the optimal control setting, we prove that our scheme converges to a stationary point sublinearly, and upgrade to linear (exponential) convergence under additional convexity assumptions. Our proof uses FM to induce an Eulerian coordinate (density-based) from a Lagrangian one (particle-based), and this also leads to certain equivalence results between the two formulations for MFGs when the Eulerian solution is sufficiently regular. Our method demonstrates promising experimental performance on MFGs in high dimensions.

## 1 Introduction

Mean-field games (MFGs) (Lasry & Lions, 2007; Huang et al., 2006) study the Nash Equilibria in games involving a continuum of indistinguishable, non-cooperative players. In an MFG, individual cost is affected by the aggregate behavior of the population $\rho$ due to interaction effects. Given the collective behavior of the population, every player seeks an optimal strategy $\hat{v}$ that minimizes their individual cost. However, as players update their strategies to $\hat{v}$, the overall population distribution also evolves to $\hat{\rho}$ induced by $\hat{v}$. A mean-field Nash equilibrium (MFNE) in this context is a state where the strategy chosen by each player is optimal with respect to the population, and the population itself is consistent with these strategies.

Mathematically, such an equilibrium can be formulated as the fixed point of an optimal control problem $\mathrm{Argmin}_{(\tilde{\rho},\tilde{v})\in C_{(\rho,v)}} \mathcal{J}(\tilde{\rho},\tilde{v};\rho)$, where the cost $\mathcal{J}$ as a functional of the players' strategy $\tilde{v}$ and individual distribution $\tilde{\rho}$ (induced by $\tilde{v}$) also involves an external population distribution $\rho$. The equilibrium is then defined as

$$(\hat{\rho},\hat{v}) \in \underset{(\tilde{\rho},\tilde{v})\in C_{(\rho,v)}}{\mathrm{Argmin}} \mathcal{J}(\tilde{\rho},\tilde{v};\rho), \quad \rho = \hat{\rho}, \tag{1}$$

that is, when the optimal control solution $\hat{\rho}$ and the population $\rho$ coincide. We refer to $(\hat{\rho},\hat{v})$ as the best response to $\rho$. The constraint set $C_{(\rho,v)}$ essentially poses a continuity equation (CE), that is, $\tilde{\rho}$ satisfies the CE associated with $\tilde{v}$. The specifics of $C_{(\rho,v)}$ and $\mathcal{J}$ will be given in the subsequent sections. In this paper, we focus on MFGs with deterministic dynamics, i.e., the agents' evolution is determined solely by their initial positions and controls, without any stochastic noise. These are also referred to as first-order MFGs, as the PDE defining the constraint set $C_{(\rho,v)}$ is of first order.

Thanks to the mean-field approximation, MFGs provide a more tractable framework for analyzing systems of a large number of interacting agents. Since their introduction, MFGs have found applications across various domains, including economics (Angiuli et al., 2023; Carmona, 2021), social sciences (Lee et al., 2021), and engineering (Djehiche et al., 2017). More recently, MFGs have

attracted growing attention due to their emerging connections with topics in machine learning, such as optimal transport (Benamou & Brenier, 2000), normalizing flows (Huang et al., 2023; Zhang & Katsoulakis, 2023), deep neural network training (E, 2017), and reinforcement learning (Mondal et al., 2022; Angiuli et al., 2023). Motivated by applications, there is a growing interest in tackling the computation of MFGs, especially in high-dimensional space.

Solving MFGs encompasses and generalizes several key problems. In the cost $\mathcal{J}(\tilde{\rho}, \tilde{v}; \rho)$, when the terms involving $\rho$ are the first variation of some potential functionals, the MFG problem (1) admits a variational formulation (Proposition 2.1). This class of problems is commonly referred to as potential MFGs or mean-field control (MFC) in the literature. On the other hand, when the cost $\mathcal{J}(\tilde{\rho}, \tilde{v}; \rho)$ is independent of $\rho$, the individual has no interactions with the population and the MFG problem (1) reduces to an individual optimal control (OC) problem. Notably, the trajectory-regularized normalizing flows for generative models (Huang et al., 2023; Xu et al., 2025) can also be formulated as an MFG. However, despite substantial recent efforts, solving high-dimensional MFGs remains challenging. Among others, one difficulty is due to their inherent fixed-point structure. In game theory, it is known that a naïve fixed-point iteration $\rho^{(k+1)} = \hat{\rho}^{(k)}$ may not converge. A widely used alternative is *fictitious play* (Cardaliaguet & Hadikhanloo, 2017) $\rho^{(k+1)} := (1 - \alpha_k)\rho^{(k)} + \alpha_k\hat{\rho}^{(k)}$, which converges for suitable $\alpha_k \in (0, 1]$ under certain assumptions (Yu et al., 2024) and has close connections to the generalized Frank-Wolfe algorithm (Lavigne & Pfeiffer, 2023). Despite its theoretical appeal, directly implementing fictitious play in high-dimensional settings becomes impractical because computing the exact best response is expensive. Nevertheless, it suggests that even when the objective changes at each step, moving toward the best response of the current step eventually leads to an MFNE.

In this work, we propose an iterative neural method for high-dimensional MFGs that, at each step, jointly updates particle trajectories in Lagrangian coordinates and trains a Flow-Matching (FM) network to parameterize the velocity field. Specifically, we propose a proximal best response scheme motivated by fictitious play. Our approach limits the search to a local neighborhood around the current state instead of seeking a global best response and averaging at each iteration as done in classical fictitious play. To implement this scheme, we introduce a concurrent particle and FM update. In each iteration, we first update a set of particles directly using the gradient-based information. Next, we train a neural flow model $v_\theta$ to match the velocity of the updated particle trajectories via a mean-squared loss as in FM (Albergo & Vanden-Eijnden, 2023; Lipman et al., 2023; Liu et al., 2023). While our setting differs from the original setup of FM, where the endpoint distributions are given (accessed via finite samples) and the goal is to interpolate to the target distribution from a source distribution, the FM step in our method has the effect of disentangling the sampled particle trajectories and ensuring that the marginal sample distributions are preserved (see Figure 1). Theoretically, we prove the convergence rate of the proposed algorithm, and the property of FM to preserve marginal distribution is used in our analysis. In practice, the FM step allows $v_\theta$ to be trained over batches of sampled trajectories and accelerates convergence.

We summarize our contribution as follows:

1. We propose a proximal fixed-point scheme that consists of a particle optimization step and a Flow Matching step concurrently in each iteration to solve high-dimensional MFGs. In particular, our scheme can solve potential MFG (including the optimal control setting) as well as general MFG by leveraging the fictitious play approach.

2. Theoretically, we prove that the particle-based proximal fixed-point scheme converges sublinearly and linearly with additional convexity assumptions. Our convergence rates hold in the optimal control setting, while some intermediate results apply to general MFGs. In particular, our proof uses a "trajectory disentangling" property of the flow matching, which also allows us to obtain certain equivalence between the Lagrangian (particle-based) and Eulerian (density-based) formulations, assuming that the Eulerian solution is sufficiently regular.

3. We apply the proposed algorithm to simulated and image datasets, including a non-potential MFG and an image-to-image translation task derived from a relaxed transport formulation within the MFG framework.

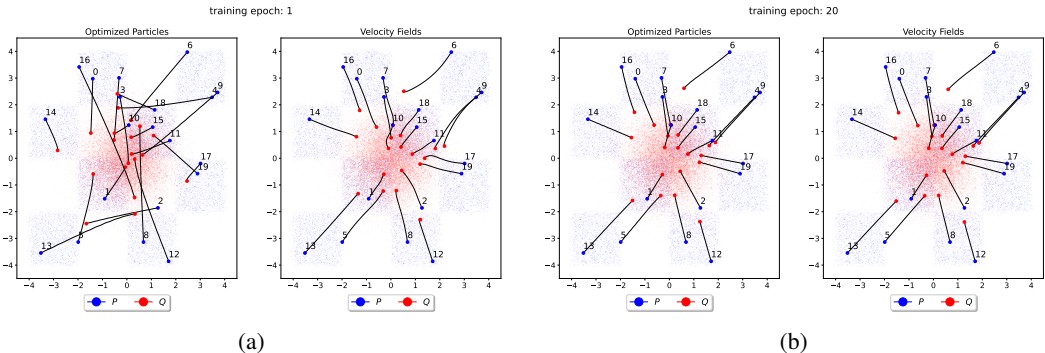

Figure 1: Illustration of the trajectory disentangle effect by Flow Matching (FM). (a) and (b) show the sample trajectories at the beginning of training and after 20 outer loops of training, respectively, where "optimized particles" stands for trajectories after a particle update, and "velocity field" shows trajectories resampled from a learned velocity field by FM, see Algorithm 1. Theoretically, we prove that FM disentangles the trajectories and reduces the dynamic cost, while leaving the interaction and terminal costs unchanged.

**Notations**  The notations are standard and detailed in Appendix A and Table 2. In particular, $L^2(\mu \otimes [0,1])$ denotes the set of time-dependent vector fields $\xi : \mathbb{R}^d \times [0,1] \to \mathbb{R}^d$ with $\int_0^1 \|\xi_t\|_\mu^2 \mathrm{d}t < \infty$, where the inner-product $\langle \xi, \eta \rangle_{\mu \otimes [0,1]} := \int_0^1 \langle \xi_t, \eta_t \rangle_\mu \mathrm{d}t$ and induces the norm $\| \cdot \|_{\mu \otimes [0,1]}$.

## 1.1 RELATED WORKS

**Fictitious play in solving MFGs**  Fictitious play, originally introduced in (Brown, 1949; 1951), is a classical algorithm in game theory. Building upon this idea, (Cardaliaguet & Hadikhanloo, 2017) applied fictitious play to MFGs using PDE analysis, proving convergence in the MFC setting, where the problem admits a variational formulation. The connection between fictitious play and the generalized Frank–Wolfe algorithm was later highlighted in (Lavigne & Pfeiffer, 2023), which leveraged this perspective to accelerate the algorithm and establish faster convergence rates in the MFC setting. This line of work was further extended to general (non-variational) MFGs in (Yu et al., 2024), achieving similar acceleration without relying on a variational structure. From a computational standpoint, the algorithms in (Lavigne & Pfeiffer, 2023; Yu et al., 2024) are implemented via the PDE formulation of the MFG, specifically using the optimality conditions of the optimization problem in (1). This leads to a coupled system of Hamilton–Jacobi–Bellman (HJB) and Fokker–Planck (FP) equations. These methods rely on mesh-based solvers, which can provide accurate numerical solutions in low-dimensional settings but struggle to scale efficiently in high-dimensional problems due to the curse of dimensionality.

Beyond PDE-based approaches, another line of work has investigated the connection between mean-field games (MFGs) and reinforcement learning (RL), using RL techniques to implement fictitious play (Perrin et al., 2020; Guo et al., 2023; Magnino et al., 2025). Most works in this direction assume a finite or discrete state space, such as a graph, which limits their scalability to continuous domains like $\mathbb{R}^d$. Although discretization via meshing is possible, it suffers from the curse of dimensionality and becomes computationally infeasible in high dimensions. Moreover, many of these studies focus on infinite-horizon settings, where the problem reduces to finding a time-invariant equilibrium. In contrast, we consider a finite-horizon setting, which requires solving a time-dependent dynamic problem and poses greater computational and theoretical challenges. Specifically, we focus on a finite time horizon rescaled to the interval $[0, 1]$.

**Flow-based methods and simulation-free methods**  To avoid discretizing the state space, recent works use neural networks to represent the trajectories of particles and define the objective directly in terms of network parameters. For example, (Ruthotto et al., 2020; Zhou et al., 2025a; Assouli et al., 2025) parameterize the value function with a neural network; the particle trajectories are then induced by the value function via ODEs. (Huang et al., 2023) uses discrete normalizing flows to model particle transitions between time steps. A common limitation of these methods is that they assume a variational formulation of MFGs, which only exists in special cases (e.g., potential MFGs or MFC). Even in those cases, the objective depends nonlinearly on the population distribution $\rho$, which is coupled with the velocity field $v$ through a PDE constraint. As a result, training requires backpropagating through PDE or ODE solvers, which is computationally expensive.

Simulation-free methods avoid solving PDEs or ODEs during the forward process of a neural network, and therefore simplify backpropagation and lead to significantly lower computational cost. These methods have recently been developed for stochastic optimal control (SOC), where the cost functions $F$ and $G$ are independent of the population distribution $\rho$, and the dynamics include stochastic noise. (Hua et al., 2025) derive analytical gradient expressions that avoid simulation and do not decouple the distribution from the control. Other works (Domingo-Enrich et al., 2024; 2025) use a decoupling approach, training the flow through control matching or adjoint equations. In comparison, our method addresses the first-order MFGs, which is a broad class of problems, and in particular, allows $F, G$ to depend on $\rho$. Conceptually, our approach leverages the Lagrangian coordinates formulation of MFGs and trains the flow velocity field by directly matching to sample velocities.

**Deep network methods for MFGs**    Beyond the works mentioned above, several other deep learning approaches have been developed for solving MFGs. (Lin et al., 2021; Gomes et al., 2023) formulate the problem as a minimax optimization and alternate updates between two variables to find a saddle point. (Chen et al., 2023a; Zhou et al., 2025b; Fouque et al., 2025) adapt actor-critic methods from reinforcement learning to the MFG setting. (Chen et al., 2023b) uses the PDE formulation of MFGs and designs two neural network modules to force the HJB and FP together. Our approach adopts flow matching techniques in modern deep generative models, and we demonstrate the efficiency of our method on high-dimensional image data.

## 2 PRELIMINARIES

### 2.1 EULERIAN AND LAGRANGIAN COORDINATES

Let $[0, 1]$ be the time interval and $\mu \in \mathcal{P}_2^r(\mathbb{R}^d)$ be a fixed initial distribution that has density, where $\mathcal{P}_2^r(\mathbb{R}^d) = \mathcal{P}_2(\mathbb{R}^d) \cap AC(\mathbb{R}^d)$. In MFGs, the distribution evolution $\tilde{\rho}$ is induced by a control flow $\tilde{v}$ via the continuity equation. We restrict our focus to distribution-control pairs $(\tilde{\rho}, \tilde{v})$ in the constraint set

$$C_{(\rho,v)} := \left\{ (\tilde{\rho}, \tilde{v}) : \partial_t \tilde{\rho} + \nabla \cdot (\tilde{\rho}\tilde{v}) = 0, \ \tilde{\rho}(\cdot, 0) = \mu, \ \tilde{\rho} \in C_\rho, \ \tilde{v}_t \in L^2(\tilde{\rho}_t) \right\}. \tag{2}$$

Here $C_\rho = AC^2(0, 1; (\mathcal{P}_2(\mathbb{R}^d), W_2))$ is the set of absolutely continuous curves in Wasserstein space (Definition B.1 in the appendix). According to (Ambrosio et al., 2008, Thm. 8.3.1), for any $\rho \in C_\rho$, there exists a Borel vector field $v$ such that $v_t \in L^2(\rho_t)$ and the pair $(\rho_t, v_t)$ satisfies the continuity equation in the sense of distributions. From a fluid dynamics perspective, $v(x, t)$ describes the motion of fluid by observing the change at a fixed spatial location over time and is known as the *Eulerian* coordinate. It is analogous to monitoring the population density at a fixed state $x$ over time. Due to its reliance on a fixed spatial grid or mesh, Eulerian simulation becomes impractical in high dimensions.

Alternatively, one may adopt a particle-based perspective by considering $X \in C_X := H^1(0, 1; L^2(\mu))$. Here, $X(x, \cdot) : [0, 1] \to \mathbb{R}^d$ represents the trajectory of an individual agent starting from position $x$ with $X(x, 0) = x$. For any a.e. $t$, the weak derivative $\partial_t \tilde{X}(\cdot, t)$ exists and is in $L^2(\mu)$. When all particles evolve under the same velocity field, the Eulerian and Lagrangian descriptions are linked by the ordinary differential equation:

$$\partial_t X(x, t) = v(X(x, t), t). \tag{3}$$

and $X$ is known as the *Lagrangian* coordinates.

Let $C_{(X,v)}$ be the constraint set where the ODE is satisfied for $\mu$-a.e. $x$ and a.e. $t$:

$$(\tilde{X}, \tilde{v}) \in C_{(X,v)} := \left\{ (\tilde{X}, \tilde{v}) : \partial_t \tilde{X}(x, t) = \tilde{v}(\tilde{X}(x, t), t), \ \tilde{X}(x, 0) = x, \ \tilde{X} \in C_X \right\}. \tag{4}$$

Notice that for $X \in C_X$, the trajectories may intersect, and there may not exist $v$ such that $(X, v) \in C_{(X,v)}$. In Section 4, we show that by flow matching, there exist $(\rho, v) \in C_{(\rho,v)}$ such that $v$ takes average on intersecting points and $\rho_t = (X_t)_{\#}\mu$. On the other hand, given $(\rho, v) \in C_{(\rho,v)}$, the ODE solution $X$ to $v$ may not always exist and may not be unique if it exists. Let $C_v$ be the set of functions $v : \mathbb{R}^d \times [0, 1] \to \mathbb{R}^d$ that are bounded and Lipschitz continuous in $x$ on every compact set $B \subset \mathbb{R}^d$. Then, by a standard existence and uniqueness theorem for ODEs (see, for example, (Ambrosio et al., 2008, Lemma 8.1.4)), there exists a unique solution $X$ such that $(X, v) \in C_{(X,v)}$ and $(X_t)_{\#}\mu = \rho_t$. In the literature, the Lagrangian formulation has also been considered in a weaker sense using probability measures on path space. We briefly discuss this perspective in Appendix B.

## 2.2 Mean-field games

Given a population distribution evolution $\rho \in C_\rho$, the individual cost of adopting strategy $(\tilde{\rho}, \tilde{v})$ depends on $\rho$. Precisely, let $F, G : \mathcal{P}_2(\mathbb{R}^d) \times \mathbb{R}^d \to \mathbb{R}$ be operators acting on the population measure. For $\rho_t \in \mathcal{P}_2(\mathbb{R}^d)$, $F[\rho_t] : \mathbb{R}^d \to \mathbb{R}$ is a function on $\mathbb{R}^d$. It assigns an interaction cost $F[\rho_t](x)$ to the individual at position $x$ at time $t$ given the population distribution $\rho_t$. The total individual cost in response to $\rho$ is defined by

$$\mathcal{J}(\tilde{\rho}, \tilde{v}; \rho) := \int_0^1 \int \left( \frac{1}{2} \|\tilde{v}(x, t)\|^2 + F[\rho_t](x) \right) \mathrm{d}\tilde{\rho}_t(x)\mathrm{d}t + \int G[\rho_1](x)\mathrm{d}\tilde{\rho}_1(x). \tag{5}$$

A pair $(\hat{\rho}, \hat{v})$ that minimizes $\mathcal{J}(\tilde{\rho}, \tilde{v}; \rho)$ over $(\tilde{\rho}, \tilde{v}) \in C_{(\rho, v)}$ is called the best response to $\rho$. The objective in an MFG is to solve (1) to find an MFNE, which is a triple $(\hat{\rho}, \hat{v}; \rho)$ such that the individual best response is consistent with the population, i.e., $\hat{\rho} = \rho$. In the objective (5), we refer to $F$ as the interaction coupling and $G$ as the terminal coupling, where $F[\rho_t]$ and $G[\rho_1]$ represent the interaction cost and terminal cost, respectively. The couplings $F, G$ are typically either local, of the form $F[\rho](x) = f(p(x))$ with $f : \mathbb{R}_{\geq 0} \to \mathbb{R}$ and $p$ being the density of $\rho$, or nonlocal of form $F[\rho](x) = (K_x * \rho)(x)$ with $K_x$ being the kernel at $x$.

When the couplings $F, G$ are $L^2$ first variations of functionals $\mathcal{F}, \mathcal{G}$ on $\mathcal{P}_2(\mathbb{R}^d)$, (5) takes the form as

$$\mathcal{J}(\tilde{\rho}, \tilde{v}; \rho) := \int_0^1 \int \left( \frac{1}{2} \|\tilde{v}(x, t)\|^2 + D\mathcal{F}[\rho_t](x) \right) \mathrm{d}\tilde{\rho}_t(x)\mathrm{d}t + \int D\mathcal{G}[\rho_1](x)\mathrm{d}\tilde{\rho}_1(x), \tag{6}$$

and the formal definition of the derivatives $D\mathcal{F}, D\mathcal{G}$ can be found in the appendix (Definition B.2). In this case, the MFG problem (1) is associated with a variational form (Lasry & Lions, 2007; Achdou et al., 2021). We summarize the result in the following proposition, and include a proof in Appendix D.1 for completeness. Note that our definition of convex functionals on $\mathcal{P}(\mathbb{R}^d)$ is with respect to affine interpolation (Definition B.3).

**Proposition 2.1.** *Let $\mathcal{F} : \mathcal{P}_2(\mathbb{R}^d) \to \mathbb{R} \cup \{+\infty\}$ and $\mathcal{G} : \mathcal{P}_2(\mathbb{R}^d) \to \mathbb{R}^d \cup \{+\infty\}$ be proper and consider the following optimization problem:*

$$\inf_{(\rho, v) \in C_{(\rho, v)}} \mathcal{J}(\rho, v) := \int_0^1 \int \frac{1}{2} \|v(x, t)\|^2 \mathrm{d}\rho_t(x)\mathrm{d}t + \int_0^1 \mathcal{F}(\rho_t)\mathrm{d}t + \mathcal{G}(\rho_1), \tag{7}$$

*If $\mathcal{F}, \mathcal{G}$ have first variations $F, G : \mathcal{P}_2(\mathbb{R}^d) \times \mathbb{R}^d \to \mathbb{R} \cup \{+\infty\}$ under $L^2$ metric, and $(\hat{\rho}, \hat{v}) \in C_{(\rho, v)}$ is the minimizer to (7), then $(\hat{\rho}, \hat{v})$ solves the fixed-point problem (1). If, in addition, $\mathcal{F}, \mathcal{G}$ are convex in $\rho$, then $(\hat{\rho}, \hat{v})$ is the minimizer to (7) if and only if $(\hat{\rho}, \hat{v})$ solves the fixed-point problem (1).*

The variational problem (7) is referred to as a potential MFG or mean-field control (MFC) problem. $\mathcal{F}$ and $\mathcal{G}$ are referred to as interaction (potential) cost and terminal (potential) cost. The variational structure facilitates the usage of many optimization algorithms and connects the MFG (1) to several important optimization problems. For example, when the interaction cost is zero, and the terminal cost is a Kullback–Leibler (KL) divergence, the MFC (7) reduces to a normalizing flow regularized by transport cost (Onken et al., 2021; Huang et al., 2023). Furthermore, since the KL divergence is convex, solving the MFG (1) is sufficient to solve the regularized normalizing flow. In the case of zero interaction cost and general terminal costs $\mathcal{G}$, the MFC (7) recovers the Jordan–Kinderlehrer–Otto (JKO) scheme (Jordan et al., 1998) for Wasserstein gradient flows, which has applications in both generative modeling (Xu et al., 2023) and solving high-dimensional kinetic equations (Huang & Wang, 2024). When $F$ and $G$ are functions on $\mathbb{R}^d$ and independent of $\rho$, the game is potential with $\mathcal{F}(\rho) = \int F(x)\mathrm{d}\rho(x)$ and $\mathcal{G}(\rho) = \int G(x)\mathrm{d}\rho(x)$ linear in $\rho$, and both (1) and (7) reduce to a single player optimal control (OC) problem. The costs $\mathcal{F}$ and $\mathcal{G}$ being linear in $\rho$ makes it convenient to approximate them with expectations.

## 2.3 Fictitious play

Fictitious play is a classical algorithm in game theory, first introduced in (Brown, 1949; 1951) and later adapted to MFGs in (Cardaliaguet & Hadikhanloo, 2017). The update rule is given by:

$$\begin{cases} (\hat{\rho}^{(k)}, \hat{v}^{(k)}) \in \underset{(\tilde{\rho}, \tilde{v}) \in C_{(\rho, v)}}{\mathrm{Argmin}} \mathcal{J}(\tilde{\rho}, \tilde{v}; \rho^{(k)}), \\ \rho^{(k+1)} = (1 - \alpha_k)\rho^{(k)} + \alpha_k\hat{\rho}^{(k)}. \end{cases} \tag{8}$$

When $\alpha_k = 1$, the update becomes a fixed-point iteration, which is shown to potentially diverge (Yu et al., 2024, Sec. 5.1). For potential MFGs, (Cardaliaguet & Hadikhanloo, 2017) proved the convergence using a diminishing weight $\alpha_k = \frac{1}{k}$. Further, (Lavigne & Pfeiffer, 2023) showed that fictitious play is equivalent to the generalized Frank-Wolfe algorithm for potential MFGs, and proved a convergence rate of $\mathcal{O}(k^{-p})$ for $\alpha_k = \frac{p}{k+p}$ (with $p > 0$). More recently, (Yu et al., 2024) extended the same convergence rate to general (non-potential) MFGs and highlighted that the best choice of the weight $\alpha_k$ depends on the local convexity of the dynamic cost.

## 3 METHOD

We first reformulate the MFG problem (1) in Lagrangian coordinates and express the objective as an expectation. We then propose a proximal fixed-point algorithm that alternates between updating the particles $X$ and the velocity field $v$. Our method utilizes optimization over sample particle trajectories, namely "particle-based", which is more scalable to high-dimensional spaces.

### 3.1 REFORMULATION IN LAGRANGIAN COORDINATES

Consider characteristic maps $X : \mathbb{R}^d \times [0,1] \to \mathbb{R}^d$ in the set $C_X$. For $\tilde{X} \in C_X$, $\tilde{X}(x, \cdot)$ denotes the trajectory of a sample particle starting from $x$. For a given $\rho \in C_\rho$, define the individual control objective in MFG as

$$\mathcal{J}(\tilde{X}; \rho) := \mathbb{E}_{x \sim \mu} \left[ \int_0^1 \left( \frac{1}{2} \|\partial_t \tilde{X}(x,t)\|^2 + F[\rho_t](\tilde{X}(x,t)) \right) dt + G[\rho_1](\tilde{X}(x,1)) \right], \quad (9)$$

and population objective in potential MFG as

$$\mathcal{J}(\tilde{X}) := \mathbb{E}_{x \sim \mu} \left[ \int_0^1 \frac{1}{2} \|\partial_t \tilde{X}(x,t)\|^2 dt \right] + \int_0^1 \mathcal{F}((\tilde{X}_t)_{\#}\mu) dt + \mathcal{G}((\tilde{X}_1)_{\#}\mu). \quad (10)$$

Using the Lagrangian coordinates, the MFG (1) can be formulated as

$$\hat{X} \in \underset{\tilde{X} \in C_X}{\text{Argmin}} \, \mathcal{J}(\tilde{X}; \rho), \quad \rho_t = \hat{\rho}_t := (\hat{X}_t)_{\#}\mu. \quad (11)$$

Similarly, the potential MFG (7) becomes

$$\min_{\tilde{X} \in C_X} \mathcal{J}(\tilde{X}). \quad (12)$$

Notice that in the reformulation, we only require $\tilde{X} \in C_X$, that is, each trajectory itself is a differentiable curve, but among each other the trajectories may intersect. In other words, for general $\tilde{X} \in C_X$, there may not exist a flow $\tilde{v}$ such that $(\tilde{X}, \tilde{v}) \in C_{(X,v)}$. In section 4, we will resolve this issue using Flow Matching (FM): FM will provide a velocity field $\tilde{v}$ such that the marginal density $\tilde{\rho}$ of $\tilde{X}$ satisfies the continuity equation associated with $\tilde{v}$. In addition, this pair $(\tilde{\rho}, \tilde{v})$ will preserve or lower the objective $\mathcal{J}$ compared to the objective on $\tilde{X}$ (Lemma 4.1). This result will further allow us to establish certain equivalence between the Eulerian and Lagrangian problems under regularity conditions.

### 3.2 PARTICLE OPTIMIZATION AND NEURAL FLOW MATCHING

A direct adaptation of fictitious play to the Lagrangian formulation (11) suggests computing the best response $\hat{X}^{(k)}$ to $\rho^{(k)}$ and then update by $X^{(k+1)} = (1 - \alpha_k)X^{(k)} + \alpha_k \hat{X}^{(k)}$ and $\rho_t^{(k+1)} = (X_t^{(k+1)})_{\#}\mu$. However, solving for the best response can be computationally expensive. Moreover, when the step size $\alpha_k$ is small, an accurate best response is often unnecessary, as the effect of any approximation error is scaled down by $\alpha_k$.

By definition, the best response provides a descent direction for the current objective $\mathcal{J}(\cdot; \rho^{(k)})$. A key observation from fictitious play is that convergence to a fixed point is possible as long as each update yields a small improvement relative to the current objective, even though the objective itself changes at each iteration. Motivated by this, we search for the proximal best response at each

iteration and propose a proximal fixed-point scheme. Specifically, given $v^{(k)}$ and a step size $\alpha_k > 0$, the approach consists of the following steps (we write in the form of integration over distributions here, and in practice implement via finite samples and time discretization):

1. Trajectory computation: for initial point $x$ which observes the distribution $\mu$, solve for its trajectory driven by $v^{(k)}$ to obtain $X^{(k)}$ such that $(X^{(k)}, v^{(k)}) \in C_{(X,v)}$.

2. Cost and particle update: update the population-dependent cost functional $\mathcal{J}(X; \rho^{(k)})$, where $\rho_t^{(k)}$ is the law of $X^{(k)}(x, t)$ with $x \sim \mu$, and then update $X$ by proximal descent

$$X^{(k+\frac{1}{2})} = \underset{X \in C_X}{\operatorname{argmin}} \ \mathcal{J}(X; \rho^{(k)}) + \frac{1}{2\alpha_k} \left( \|X - X^{(k)}\|_{\mu \otimes [0,1]}^2 + \|X_1 - X_1^{(k)}\|_\mu^2 \right). \quad (13)$$

3. Flow matching update: update $v$ by regression

$$v^{(k+1)} := \underset{v}{\operatorname{argmin}} \, \mathbb{E}_{x \sim \mu} \left[ \int_0^1 \left\| v(X^{(k+\frac{1}{2})}(x, t), t) - \partial_t X^{(k+\frac{1}{2})}(x, t) \right\|^2 \mathrm{d}t \right]. \quad (14)$$

In Lemmas 4.4 and 4.5, we prove that both the particle update (13) and the flow matching update (14) are well-defined. When $X^{(k)}$ is not a stationary point, the particle update yields a decrease in the objective function. More importantly, if $X^{(k+\frac{1}{2})}$ is not induced by a velocity field $v$, then the subsequent flow matching and resampling steps also result in a decay of the objective. Based on this observation, we perform the flow matching update less frequently in practice, as it contributes to improvement primarily when particle trajectories intersect.

In practice, the proximal descent update of particles (13) is implemented by several iterations of gradient descent. To derive the descent direction, consider a smooth perturbation $Y : \mathbb{R}^d \times [0, T] \to \mathbb{R}^d$ satisfying $Y(x, 0) = 0$. For sufficiently small $\epsilon$, a first-order expansion of $\mathcal{J}(X + \epsilon Y; \rho)$ at $\epsilon$ yields $\mathcal{J}(X; \rho) + \epsilon \left( \int_0^1 \langle Y_t, -(\partial_{tt} X)_t + \nabla F[\rho_t](X_t) \rangle_\mu \, \mathrm{d}t + \langle Y_1, (\partial_t X)_1 + \nabla G[\rho_1](X_1) \rangle_\mu \right)$. The coefficient of $\epsilon$ characterizes the descent direction. In particular, at intermediate times $t \in (0, 1)$, the particles evolve along the direction $(\partial_{tt} X)_t - \nabla F[\rho_t](X_t)$, while at the terminal time, they follow $-((\partial_t X)_1 + \nabla G[\rho_1](X_1))$. Consequently, each cost update requires evaluating the functionals $F[\rho_t]$ and $G[\rho_1]$, together with their (sub)gradients. The specific form of the update depends on the choice of $F$ and $G$. We provide two representative examples in Examples 3.1 and 3.2.

**Example 3.1.** When $\mathcal{G}(\rho) = \mathrm{KL}(\rho \| \nu)$, we have $G[\rho] = \log \frac{\mathrm{d}\rho}{\mathrm{d}\nu}$. In our particle-based method, we are to update the cost $G[\rho]$ when the distribution $\rho$ (and possibly also $\nu$) is given by finite samples. An estimator of $G[\rho]$ which provides its (sub)gradient at any query point $x$ can fulfill the need. In our experiments, we use a neural network classifier to approximate $G[\rho]$, and the gradient is computed via auto-differentiation.

**Example 3.2.** A nonlocal coupling is of the form $F[\rho](x) = \int w(x, y) \mathrm{d}\rho(y)$ where $w : \mathbb{R}^d \times \mathbb{R}^d \to \mathbb{R}$ is a kernel function. If $w(x, y)$ is differentiable in $x$, then $\nabla F[\rho](x) = \int \nabla_x w(x, y) \mathrm{d}\rho(y)$. Both $F[\rho]$ and $\nabla F[\rho]$ can be approximated via empirical averages over particles.

For the flow matching update, we parametrize $v$ by a neural network $v_\theta$ and (14) is implemented using standard stochastic optimization schemes such as Adam. The complete procedure is summarized in Algorithm 1, and more details are given in Appendix C.

## 4 THEORY

We begin by noting that the original problem is posed in the Eulerian formulation (1), while our algorithm operates in the Lagrangian formulation (11). In this section, we first derive a certain equivalence between the two formulations in Theorem 4.3. In Section 4.2, we analyze the convergence of the proposed scheme (13)(14), proving its descent property under for general MFGs and its convergence rates in the optimal control setting.

### 4.1 EQUIVALENCE BETWEEN EULERIAN AND LAGRANGIAN FORMULATIONS

To prove Theorem 4.3, we first relate $\tilde{X} \in C_X$ to a corresponding pair $(\tilde{\rho}, \tilde{v}) \in C_{(\rho,v)}$ and compare their associated costs. Lemma 4.1 shows that for any $\tilde{X} \in C_X$, there is a unique pair $(\tilde{\rho}, \tilde{v}) \in C_{(\rho,v)}$

---

**Algorithm 1:** Particle-based Flow-Matching for Mean-Field Games

---

1 Initialize the velocity field $v_\theta$
2 **for** $k = 1, 2, \cdots, K$ **do**
3      Trajectory sampling: sample a batch of the initial point $\{X_{i,0}\}_{i=1}^{B}$ and compute trajectories $X^B := \{X_{i,t_j}, 0 \le j \le m\}_{i=1}^{B}$ by intergrating the ODE using the velocity field $v_\theta$;
4      Cost and particle update: update $X^B$ using cost functionals $F$ and $G$ by (23) for $L_1$ steps; every $L_0$ steps, update $F$ and $G$ from the current trajectories $X^B$;
5      Flow matching: update $v_\theta$ to match the trajectories $X^B$ by minimizing the loss (24) for $L_2$ steps;
6 **end**

---

obtained by flow matching. In addition, the cost of $\tilde{X}$ is bounded below by the cost of $(\tilde{\rho}, \tilde{v})$. In the other direction, Lemma 4.2 establishes that the equality holds when $\tilde{X}$ is induced by a velocity field $\tilde{v} \in C_v$. These two lemmas will also be used in the convergence analysis in Section 4.2, and specifically, to prove Lemma 4.5. The proofs of Lemmas 4.1 and 4.2 are given in Appendices D.2 and D.3, respectively.

**Lemma 4.1.** *Let $\tilde{X} \in C_X$ and set $\tilde{v} = \mathrm{argmin}_v \, \mathbb{E}_{x \sim \mu} \left[ \int_0^1 \|v(\tilde{X}(x,t), t) - \partial_t \tilde{X}(x,t)\|^2 \mathrm{d}t \right]$, $\tilde{\rho}_t := (\tilde{X}_t)_\# \mu$. Then, $\tilde{\rho} \in C_\rho$, $(\tilde{\rho}, \tilde{v}) \in C_{(\rho, v)}$, and $\mathcal{J}(\tilde{\rho}, \tilde{v}; \rho) \le \mathcal{J}(\tilde{X}; \rho)$ for any $\rho \in C_\rho$.*

**Lemma 4.2.** *If $(\tilde{\rho}, \tilde{v}) \in C_{(\rho, v)}$ and $\tilde{v} \in C_v$, then there exists $\tilde{X} \in C_X$ satisfying $(\tilde{X}, \tilde{v}) \in C_{X,v}$ and for any $\rho \in C_\rho$, $\mathcal{J}(\tilde{X}; \rho) = \mathcal{J}(\tilde{\rho}, \tilde{v}; \rho)$.*

As a direct result of Lemmas 4.1 and 4.2, the following theorem shows the relationship between Eulerian coordinates and the Lagrangian coordinates. Specifically, under certain regularity conditions, we show that for general MFGs, an Eulerian solution induces a Lagrangian solution; For potential MFGs, the optimal value of the Eulerian and Lagrangian formulation coincides, and a Lagrangian solution also induces an Eulerian solution by flow matching. The proof is in Appendix D.4.

**Theorem 4.3.** *(i) Suppose $(\rho^*, v^*)$ is a solution to the MFG (1) and $v^* \in C_v$, then $v^*$ induces characteristics $X^* \in C_X$ that is a solution to (11).*

*(ii) Suppose the potential MFG (7) admits a solution $(\rho^*, v^*)$ with $v^* \in C_v$. Then $v^*$ induces characteristics $X^* \in C_X$ that is a solution to (12) and $\mathcal{J}(X^*) = \mathcal{J}(\rho^*, v^*) =: J^*$. In addition, for any other $X \in C_X$ that is also a solution to (12), let $(\rho, v)$ be the flow-matched Eulerian representation of $X$ as in Lemma 4.1, then $(\rho, v)$ is a solution to (7), and in this case $J(X) = J(\rho, v) = J^*$.*

### 4.2 CONVERGENCE RATE OF THE PROXIMAL FIXED-POINT SCHEME

In our scheme, the particle update (13) reduces the total cost; The flow-matching update (14) regularizes the trajectories to reduce the dynamic cost while keeping the interaction and terminal costs unchanged. As a result, the three steps together yield a decay in the objective value, as stated in the following two lemmas. Proofs are provided in Appendix D.5 and D.6.

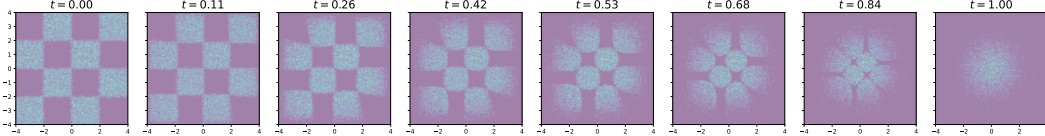

Figure 2: Intermediate distributions between $4 \times 4$ checkerboard and an isotropic Gaussian solved by the proposed algorithm.

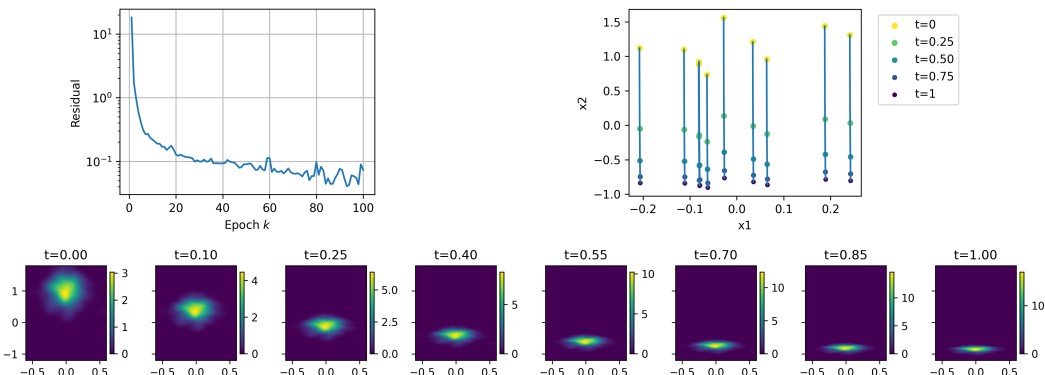

Figure 3: Non-potential MFG in Section 5.2. (Top left) Residual defined in (57) versus the loop index $k$. (Top right) Test trajectories computed from the learned $v_\theta$. Players move toward the target line $x_2 = -1$, adapting their speed to reduce the costs. (Bottom) Estimated density evolution (from test trajectories). The population tends to concentrate along $x_2$ and spreads along $x_1$ due to asymmetric interaction.

**Lemma 4.4.** *Assume that $F[\rho_t^{(k)}], G[\rho_1^{(k)}]$ are proper and L-smooth in $\mathbb{R}^d$ for any $t \in [0,1]$. Then for any $0 < \alpha_k < \frac{1}{L}$, the update scheme (13) admits a unique solution $X^{(k+\frac{1}{2})} \in C_X$ and*

$$\mathcal{J}(X^{(k+\frac{1}{2})}; \rho^{(k)}) \leq \mathcal{J}(X^{(k)}; \rho^{(k)}) - \frac{1}{2\alpha_k} \left( \|X^{(k+\frac{1}{2})} - X^{(k)}\|^2_{\mu \otimes [0,1]} + \|X_1^{(k+\frac{1}{2})} - X_1^{(k)}\|^2_\mu \right). \tag{15}$$

**Lemma 4.5.** *If $X^{(k+\frac{1}{2})} \in C_X$, set $\rho_t^{(k+1)} := (X_t^{(k+\frac{1}{2})})_{\#}\mu$. Then, $\rho^{(k+1)} \in C_\rho$ and there exists $v^{(k+1)}$ solves (14) and $v_t^{(k+1)}$ is unique upto a $\rho_t$-zero measure set. In addition, if $v^{(k+1)} \in C_v$, then there exist $X^{(k+1)} \in C_X$ such that $(X^{(k+1)}, v^{(k+1)}) \in C_{(X,v)}$, and for all such $X^{(k+1)}$, $\mathcal{J}(X^{(k+1)}; \rho) \leq \mathcal{J}(X^{(k+\frac{1}{2})}; \rho)$.*

For general MFGs, establishing how the descent property leads to convergence to a fixed point is nontrivial and remains an open question for future investigation. In this paper, we show that the proposed scheme achieves a sublinear convergence rate for optimal control problems (that is, where $F, G$ are independent of $\rho$) and a linear convergence rate when $F, G$ are additionally strongly convex on $\mathbb{R}^d$. The result is stated below, and the proof is in Appendix D.7.

**Theorem 4.6.** *Let the sequence $X^{(k)}, v^{(k)}$ generated by update scheme (13)(14) with $0 < \alpha_k = \alpha < \frac{1}{L}$, and assume $v^{(k)} \in C_v$ for all $k$. If $F, G$ are independent of $\rho$ and are proper and L-smooth functions in $\mathbb{R}^d$, and $\mathcal{J}(X) \geq \underline{\mathcal{J}}$ for any $X \in C_X$, then $X^{(k)}$ satisfies*

$$\min_{k \leq K} \left\{ \|X^{(k+\frac{1}{2})} - X^{(k)}\|^2_{\mu \otimes [0,1]} + \|X_1^{(k+\frac{1}{2})} - X_1^{(k)}\|^2_\mu \right\} \leq 2\alpha(\mathcal{J}(X^{(0)}) - \underline{\mathcal{J}})/K, \tag{16}$$

*If in addition, $F$ and $G$ are $\lambda$-convex ($\lambda > 0$), and there exist a solution $(X^*, \rho^*)$ to (11), then the sequence $X^{(k)}$ generated by update scheme (13) and $X^{(k)} = X^{(k+\frac{1}{2})}$ with $0 < \alpha_k = \alpha < \frac{1}{L}$ satisfies*

$$\|X^{(k)} - X^*\|^2_{\mu \otimes [0,1]} + \|X_1^{(k)} - X_1^*\|^2_\mu \leq \frac{1}{(1+2\lambda\alpha)^k} \left( \|X^{(0)} - X^*\|^2_{\mu \otimes [0,1]} + \|X_1^{(0)} - X_1^*\|^2_\mu \right). \tag{17}$$

## 5 EXPERIMENTS

We consider two types of MFGs: Section 5.1 and 5.3 compute the transport between two distributions via an MFG formulation, and Section 5.2 considers a non-potential MFG. Additional experimental details are provided in Appendix E, and code can be found at `https://github.com/jiajia-yu/mfg_flow_matching`.

### 5.1 TOY EXAMPLE

We are to learn a transport in $\mathbb{R}^2$ from a $4 \times 4$ checkerboard distribution to an isotropic Gaussian. In this case, $F[\rho] = 0$ and $G[\rho] = \log \frac{d\rho}{d\nu}$ as in Example 3.1, where $\nu$ is $\mathcal{N}(0, I)$. Figure 2 visualizes the evolution of the distribution, where the learned transport map continuously interpolates between the source and target densities.

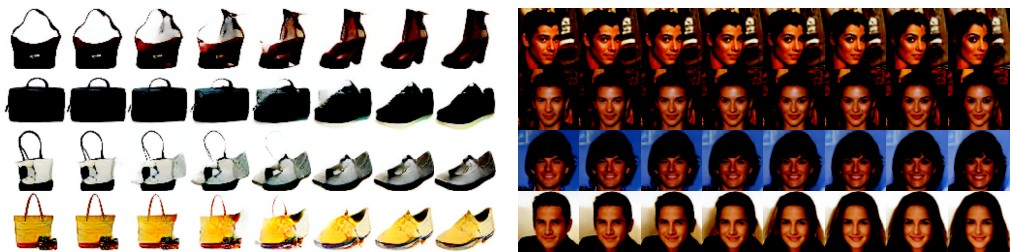

Figure 4: Randomly selected four OT trajectories for image-to-image translation of handbags → shoes (left) and CelebA male → CelebA female (right).

Table 1: FID of translated images of our method and baselines. Lower FID is better. Results for Disco GAN, Cycle GAN, and NOT are quoted from Korotin et al. (2023), and results for Q-Flow are from Xu et al. (2025).

|  | **Ours** | Q-Flow* | OT-CFM | SI | Rectified Flow | SB-CFM | Disco GAN* | Cycle GAN* | NOT* |
|---|---|---|---|---|---|---|---|---|---|
| Handbag → Shoes | 12.44 | 12.34 | 13.01 | 15.87 | 13.91 | 12.70 | 22.42 | 16.00 | 13.77 |
| Male → Female | 9.68 | 9.66 | 12.88 | 16.39 | 14.01 | 11.55 | 35.64 | 17.74 | 13.23 |

## 5.2 Non-potential MFG

We consider an MFG on $\mathbb{R}^2$ with interaction cost $F[\rho](x,t) = \lambda_F \int w(x,y)\mathrm{d}\rho(y)$, where $w(x,y) = \exp(a^\top(x-y))$ and $a \in \mathbb{R}^2$ is a fixed non-zero vector. The is the smoothing coupling as in Example 3.2. Since $w$ is asymmetric, the game is non-potential. The terminal cost is $G(x) = \lambda_G(a^\top x - c)^2$, where $c \in \mathbb{R}$, encouraging players to move toward the hyperplane $a^\top x = c$. The terminal cost is independent of $\rho$, and its gradient has a closed-form evaluation. More details can be found in Appendix E.2. Figure 3 (top left) shows that the algorithm converges to an equilibrium state with a residual of $10^{-1}$. The learned flow is visualized in the top right plot on test trajectories. The bottom plot shows the density evolution. The results show that players move toward the terminal line $a^\top x = c$ to reduce terminal cost, and as players evolve, the population compresses along the direction of $a$ and spreads along $a^\perp$, making $x - y$ nearly orthogonal to $a$. This density evolution aligns with the strategy of matching population pace while progressing toward $a^\top x = c$.

## 5.3 Image-to-image translation

We formulate the image-to-image translation task as a transport from source to target distributions under the MFG framework, by employing a KL divergence terminal cost, same as in the toy example. We consider two tasks: (i) Handbags → Shoes (Yu & Grauman, 2014; Zhu et al., 2016), and (ii) CelebA male → CelebA female (Liu et al., 2015). In each case, the objective is to generate target-domain images conditioned on source-domain inputs. Following Rombach et al. (2022), we first train a deep variational autoencoder (VAE) to embed images into a latent space, and all methods learn transport dynamics in this latent space. We evaluate performance using the Fréchet Inception Distance (FID) (Heusel et al., 2017); implementation details and baselines are provided in Appendix E.3. As reported in Table 1, our method achieves FID scores comparable to Q-Flow and outperforms other baselines. Figure 4 shows representative translation trajectories, illustrating smooth transitions with improved color consistency and fewer visual artifacts.

## 6 Discussion

The work can be extended in several future directions. First, the particle-based method incurs significant memory costs when the number of trajectories and the number of time steps are large. Second, while our framework applies to general MFGs without requiring a variational formulation, the theoretical convergence guarantees currently only cover the optimal control setting. Third, the convergence analysis considers the idealized proximal fixed point scheme, and it would be useful to develop a more complete analysis that accounts for practical sources of error, such as sampling, finite difference approximations, and flow matching inaccuracies. Finally, it would be useful to extend the method to more datasets and real applications.

ACKNOWLEDGMENTS

The authors thank Chen Xu and Edward Chen for their help with numerical experiments. The project was supported by the Simons Foundation (grant ID: MPS-MODL-00814643). XC was also partially supported by NSF DMS-2237842. The work of JL and YX was partially funded by NSF DMS-2134037, CMMI-2112533, and the Coca-Cola Foundation.

ETHICS STATEMENT

We affirm that this research complies with the ICLR Code of Ethics. No conflicts of interest are associated with this work.

REPRODUCIBILITY STATEMENT

The source code for the experiments is publicly available. The authors clarified details of all experiments for reproducibility in the Numerical Experiments section or the Appendix.

THE USAGE OF LARGE LANGUAGE MODELS (LLMS)

The authors used a large language model (ChatGPT, based on GPT-4 architecture) to assist with revising the writing of this paper. The model was used solely for improving the clarity, grammar, and fluency of the text, without contributing to the conceptual, methodological, or experimental content.

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

## A    Notations

We summarize the notations in Table 2.

A hat $\hat{\cdot}$ denotes best response quantities. Let $\mathcal{P}_2(\mathbb{R}^d)$ be the space of Borel probability measures with finite second moments. We use both the $L^2$ and Wasserstein-2 ($W_2$) metrics on $\mathcal{P}_2(\mathbb{R}^d)$, specifying which is used when needed. For $\mu \in \mathcal{P}_2(\mathbb{R}^d)$, we define $L^2(\mu)$ as the Hilbert space of vector fields $\xi : \mathbb{R}^d \to \mathbb{R}^d$ with $\int \|\xi(x)\|^2 \mathrm{d}\mu(x) < \infty$, inner product $\langle \xi, \eta \rangle_\mu := \int \xi(x) \cdot \eta(x) \mathrm{d}\mu(x)$, and the induced norm $\| \cdot \|_\mu$. The symbol $\rho$ denote a static distribution in $\mathcal{P}_2(\mathbb{R}^d)$ or a time-dependent mapping $[0, 1] \to \mathcal{P}_2(\mathbb{R}^d)$, with $\rho_t$ denoting the distribution at time $t$. Similar conventions apply to $v$ and $X$, which may refer to functions $\mathbb{R}^d \to \mathbb{R}^d$ or time-dependent maps $\mathbb{R}^d \times [0, 1] \to \mathbb{R}^d$. Constraint sets are denoted by $C_.$, where the subscript indicates the constrained variable. Objective functionals are denoted by $\mathcal{J}$. For instance, $\mathcal{J}(\tilde{\rho}, \tilde{v}; \rho)$ evaluates the cost of a distribution-velocity flow under a fixed reference flow $\rho$, and $\mathcal{J}(X; \rho)$ evaluates the cost of a characteristic map under a fixed reference flow $\rho$.

Table 2: Summary of Notations

| Notation | Description |
| --- | --- |
| $\mathcal{P}_2(\mathbb{R}^d)$ | Set of Borel probability measures on $\mathbb{R}^d$ with finite second moment. |
| $L^2(\mu)$ | Space of functions $\xi : \mathbb{R}^d \to \mathbb{R}^d$ such that $\int \|\xi(x)\|^2 \mathrm{d}\mu(x) < \infty$. |
| $\langle \xi, \eta \rangle_\mu$ | Inner product on $L^2(\mu)$ defined as $\int \langle \xi(x), \eta(x) \rangle \mathrm{d}\mu(x)$. |
| $\|\xi\|_\mu$ | Norm on $L^2(\mu)$ defined as $(\langle \xi, \xi \rangle_\mu)^{\frac{1}{2}}$. |
| $L^2(\mu \otimes [0, 1])$ | Space of functions $\xi : \mathbb{R}^d \times [0, 1] \to \mathbb{R}^d$ with finite $\mu \otimes$ Lebesgue-norm. |

*(continued on next page)*

| Notation | Description |
|---|---|
| $\langle \xi, \eta \rangle_{\mu \otimes [0,1]}$ | Inner product on $L^2(\mu \otimes [0,1])$ defined as $\int_0^1 \langle \xi_t, \eta_t \rangle_\mu \, \mathrm{d}t$. |
| $\|\xi\|_{\mu \otimes [0,1]}$ | Norm on $L^2(\mu \otimes [0,1])$ defined as $\left( \langle \xi, \xi \rangle_{\mu \otimes [0,1]} \right)^{\frac{1}{2}}$. |
| $\mu$ | Fixed initial distribution in $\mathcal{P}_2(\mathbb{R}^d)$. |
| $\rho$ | Population distribution. Depending on context, $\rho \in \mathcal{P}_2(\mathbb{R}^d)$ or $\rho : [0,1] \to \mathcal{P}_2(\mathbb{R}^d), \rho_t = \rho(t)$. |
| $v$ | Control function, velocity field. Depending on context, $v : \mathbb{R}^d \to \mathbb{R}^d$ or $v : [0,1] \to L^2(\mathbb{R}^d; \mathbb{R}^d), v_t = v(t)$. |
| $X$ | Characteristic map. Depending on context, $X : \mathbb{R}^d \to \mathbb{R}^d$ or $X : [0,1] \to L^2(\mu), X_t = X(t)$. |
| $m$ | Momentum, vector-valued Radon measure $m = \rho v$. |
| $C_\rho$ | Space of absolutely continuous curves $AC^2(0, 1; (\mathcal{P}_2(\mathbb{R}^d), W_2))$ with finite quadratic energy. |
| $C_X$ | Set of maps $X : \mathbb{R}^d \times [0,1] \to \mathbb{R}^d$ such that $X(x, \cdot) \in AC^2(0, 1; \mathbb{R}^d)$ for $\mu$-a.e. $x$. |
| $C_v$ | Set of functions $v : \mathbb{R}^d \times [0,1] \to \mathbb{R}^d$ that are bounded and Lipschitz continuous in $x$ on every compact set $B \subset \mathbb{R}^d$ |
| $C_{(\rho,v)}$ | Feasible set of pairs of distribution flow and velocity field $(\rho, v)$, defined in (2). |
| $C_{(X,v)}$ | Feasible set of pairs of characteristic map and velocity field $(X, v)$, defined in (4). |
| $C_{(\rho,m)}$ | Feasible set of pairs of distribution flow and vector-valued Radon measure flow $(\rho, m)$ defined in (25). |
| $\mathcal{R}(\rho, v)$ | Dynamic cost of $(\rho, v) \in C_{(\rho,v)}$. $\mathcal{R}(\rho, v) := \int_0^1 \int \|v_t(x)\|^2 \mathrm{d}\rho_t(x)$. |
| $\mathcal{R}(X)$ | Dynamic cost of $X \in C_X$. $\mathcal{R}(X) := \int_0^1 \int \|\partial_t X(x, t)\|^2 \mathrm{d}\mu(x)$. |
| $\mathcal{R}(\rho, m)$ | Dynamic cost of $(\rho, m) \in C_{(\rho,m)}$. $\mathcal{R}(\rho, m) := \int_0^1 \int \|\frac{\mathrm{d}m_t}{\mathrm{d}\rho_t}\|^2 \mathrm{d}\rho_t(x)$. |
| $F$ | Interaction cost coupling, $\mathcal{P}(\mathbb{R}^d) \times \mathbb{R}^d \to \mathbb{R}, (\rho, x) \mapsto F[\rho](x)$. See Exs. 3.1, 3.2. |
| $G$ | Terminal cost coupling, similar to $F$. |
| $\mathcal{J}(\rho, v; \tilde{\rho})$ | Individual optimal control objective on $C_{(\rho,v)}$ for given $\tilde{\rho} \in C_\rho$, defined in (5). |
| $\mathcal{J}(X; \tilde{\rho})$ | Individual optimal control objective on $C_X$ for given $\tilde{\rho} \in C_\rho$, defined in (9). |
| $\mathcal{J}(\rho, m; \tilde{\rho})$ | Individual optimal control objective on $C_{(\rho,m)}$ for given $\tilde{\rho} \in C_\rho$, defined in (27). |
| $\mathcal{F}, \mathcal{G}$ | Interaction and terminal potentials, $\mathcal{P}_2(\mathbb{R}^d) \to \mathbb{R}$. |
| $\mathcal{J}(\rho, v)$ | MFC/potential MFG objective on $C_{(\rho,v)}$, defined in (7). |
| $\mathcal{J}(X)$ | MFC/potential MFG objective on $C_X$, defined in (10). |
| $\mathcal{J}(\rho, m)$ | MFC/potential MFG objective on $C_{(\rho,m)}$, $\mathcal{J}(\rho, m) := \mathcal{R}(\rho, m) + \int \mathcal{F}(\rho_t)\mathrm{d}t + \mathcal{G}(\rho_1)$. |
| $m, \Delta t, t_j$ | $m$ is number of time steps, $\Delta t = \frac{1}{m}$ and $t_j = j\Delta t$. |
| $X_{i,t_j}$ | Approximate location of particle initialized at $x_i$ at time $t_j$, i.e., $X_{i,t_j} \approx X(x_i, t_j)$. |
| $\alpha_k$ | Weight for fictitious play in (8) or the proximal step-size in (13). |
| $\beta$ | Step-size for particle update in (23). |
| $K$ | Total number of outer loops in Algorithm 1. |
| $B$ | Number of samples per outer loop. |
| $L_0$ | Frequency of cost update. |

*(continued on next page)*

| Notation | Description |
|---|---|
| $L_1, L_2$ | Number of particle update iterations and flow neural network update iterations, respectively. |

# B  ADDITIONAL PRELIMINARIES

**Absolutely continuous curves**   In our setup, we ask $\rho$ to be an absolutely continuous curve in $(\mathcal{P}_2(\mathbb{R}), W_2)$ and $X(x, \cdot)$ an absolutely continuous curve in $\mathbb{R}$ with Euclidean distance for $\mu$-a.e. $x$. We recall the concept of absolutely continuous curves as defined in (Ambrosio et al., 2008, Def. 1.1.1).

**Definition B.1** (Absolutely continuous curves). Let $(\mathcal{X}, d)$ be a complete metric space and let $l : (0, 1) \to \mathcal{X}$ be a curve. We say that $l \in AC^2(0, 1; \mathcal{X})$ if there exists a function $m \in L^2(0, 1)$ such that

$$d(l(s), l(t)) \le \int_s^t m(r)\mathrm{d}r, \quad \forall 0 < s \le t < b. \tag{18}$$

By (Ambrosio et al., 2008, Thm. 1.1.2), for any $l \in AC^2(0, 1; \mathcal{X})$, the limit

$$|l'|(t) := \lim_{s \to t} \frac{d(l(s), l(t))}{|s - t|} \tag{19}$$

exists for a.e. $t \in (0, 1)$. In addition, $|l'| \in L^2(0, 1)$ and $|l'|$ is an admissible integrand of (18) and for any $m$ satisfying (18), $|l'|(t) \le m(t)$ for a.e. $t \in (0, 1)$. $|l'|$ is called the metric derivative.

**First variation in probability space**   In Proposition 2.1, we show that when $F, G$ are first variations of some $\mathcal{F}, \mathcal{G}$, the MFG (1) is associated to an optimization problem (7). The first variation in probability space is also called the Lions derivative. The main difference between it and Fréchet derivative is that the perturbation is required to have zero measure.

**Definition B.2** (First variation (Cardaliaguet & Hadikhanloo, 2017)). Let $\mathcal{F} : \mathcal{P}_2(\mathbb{R}^d) \to \mathbb{R}$, we say that $D\mathcal{F} : \mathcal{P}_2(\mathbb{R}^d) \times \mathbb{R}^d \to \mathbb{R}$ is the first variation of $\mathcal{F}$ if for any $\rho, \rho' \in \mathcal{P}_2(\mathbb{R}^d)$,

$$\lim_{s \to 0} \frac{1}{s} \left( \mathcal{F}((1-s)\rho + s\rho') - \mathcal{F}(\rho) \right) = \int D\mathcal{F}[\rho](x)\mathrm{d}(\rho' - \rho)(x). \tag{20}$$

we say that $D\mathcal{F} : \mathcal{P}_2(\mathbb{R}^d) \times \mathbb{R}^d \to \mathbb{R}$ is the first-order subdifferential of $\mathcal{F}$ if for any $\rho, \rho' \in \mathcal{P}_2(\mathbb{R}^d)$,

$$\mathcal{F}((1-s)\rho + s\rho') - \mathcal{F}(\rho) \ge s \int D\mathcal{F}[\rho](x)\mathrm{d}(\rho' - \rho)(x). \tag{21}$$

Since $D\mathcal{F}$ is defined up to an additive constant, we assume that

$$\int D\mathcal{F}[\rho](x)\mathrm{d}\rho(x) = 0, \quad \forall \rho \in \mathcal{P}_2(\mathbb{R}^d). \tag{22}$$

**Convexity of cost functionals on $\mathcal{P}_2(\mathbb{R}^d)$**

**Definition B.3.** We say $\mathcal{F} : \mathcal{P}_2(\mathbb{R}^d) \to \mathbb{R}$ is convex if

$$\mathcal{F}((1-\delta)\rho^0 + \delta\rho^1) \le (1-\delta)\mathcal{F}(\rho^0) + \delta\mathcal{F}(\rho^1)$$

for any $\delta \in [0, 1], \rho^0, \rho^1 \in \mathcal{P}_2(\mathbb{R}^d)$.

**Comparison with path-space formulations**   In the studies of optimal transport and optimal control, a useful extension of the classical ODE flow is via measures on the path space (Superposition Principle (Ambrosio, 2008; Ambrosio et al., 2008)), which, e.g., allows branching of trajectories that follow the velocity field $v$ in a weaker sense Lisini (2006); Cavagnari et al. (2022). In this paper, we parametrize the velocity field $v$ by a neural network, which in practice usually gives a regular $v$. Thus, we focus on when $v$ is regular, and specifically when the solution to MFG (1) has $v \in C_v$.

We also comment on the results of Theorem 4.3 in comparison to the equivalence results (between the Lagrangian and Eulerian formulations) derived under the path space. For control problems, such an equivalence has been previously studied in (Cavagnari et al., 2022) by considering distributions on path space, and their Lagrangian formulation is in a weak sense. In contrast, our framework is motivated by neural network methodology, and we focus our theory on classical ODE flows where trajectories from $\mu$-a.e. $x$ are well-defined, which is possible when assuming the MFG solution has a regular $v^*$.

## C  ALGORITHM DETAILS

We provide details for Algorithm 1 to implement the proposed method in Section 3.

The velocity field $v$ is parameterized by a neural network $v_\theta$, and $X$ is represented by time-discretized sampled trajectories $X(x, \cdot)$. Specifically, let $\Delta t = 1/m$ and define the discrete time grid by $t_j = j \Delta t$ for $j = 1, \ldots, m$. The trajectory of the $i$-th particle is denoted as $\{X_{i,t_j}\}_{j=1}^m$, where $X_{i,t_j} \approx X(x_i, t_j)$, and $x_i$ is the initial point drawn from a distribution $\mu$.

**Initial velocity field**  Before the outer loop, we need to specify an initial value for $v_\theta$. When there is prior knowledge, we can use it to prescribe the initial velocity field. For example, $v(x, t) = 0$ for all $(x, t)$, when appropriate. We adopt this initialization for the non-potential MFG example in Section 5.2. Alternatively, one may first assign a target point to each initial sample and construct trajectories via linear interpolation between the initial and target points. The network $v_\theta$ can then be initialized through a flow-matching step on these interpolated trajectories. We use this strategy for the transport tasks in Sections 5.1 and 5.3, where the target points are sampled from the desired target distributions. This initialization provides a favorable starting point, as it produces a terminal distribution that is already close to the desired target distribution.

**Loop iterations**  Below are the details for the loop iterations:

1. Trajectory sampling.

   We first sample a batch $B$ of initial points $\{x_i\}_{i=1}^B$. In our experiments, $x_i \sim \mu$ is sampled on the fly if $\mu$ is analytically known; When only finite samples of $\mu$ are given (in a training set), we sample $x_i$ from the dataset. Given an ODE velocity field $v_\theta$ and an initial point $x_i$, we numerically integrate the trajectory $X(x_i, t)$ using standard schemes, such as fourth-order Runge–Kutta (RK4).

2. Cost update. The empirical distribution of the particle locations $\{X_{i,t_j}\}_{i=1}^B$ approximates the density $\rho_{t_j}$ at time $t_j$, and is used to estimate the coupling functionals $F[\rho_{t_j}]$ and $G[\rho_{t_m}] : \mathbb{R}^d \to \mathbb{R}$, along with their corresponding (sub)gradients. The resulting estimators depend on the specific form of the coupling.

   For Ex 3.1, when the classifier is updated frequently, the particle distribution changes gradually, so the classifier can be trained for fewer steps at each update.

   For Ex 3.2, the functional $F$ and its gradient admit representations in terms of expectations, which can be approximated using empirical samples. In contrast, $G$ and its gradient have closed-form expressions and can therefore be evaluated directly.

   Particle update. In practice, we update particle trajectories $X^B = \{X_{i,t_j}, 0 \le j \le m\}_i$ for $L_1$ steps by:

   $$X_{i,t_j} \leftarrow \begin{cases} X_{i,0}, & j = 0, \\ X_{i,t_j} - \beta \Delta t \left( -(D_{tt} X_i)_{t_j} + \nabla F[\rho_{t_j}](X_{i,t_j}) \right), & j = 1, 2, \cdots, m-1, \\ X_{i,t_m} - \beta \left( (D_t X_i)_{t_m} + \nabla G[\rho_{t_m}](X_{i,t_m}) \right), & j = m. \end{cases} \quad (23)$$

   Here $\leftarrow$ indicates that $X_{i,t_j}$ is updated by the right-hand side expression, $\beta$ is the step size, and $D_{tt}$ and $D_t$ are finite-difference approximations of $\partial_{tt}$ and $\partial_t$, respectively.

3. Flow matching update.

   The flow matching update trains the velocity network $v_\theta$ to match the empirical velocities of particle trajectories $X^B$, which is closely related to flow matching methods (Albergo &

Vanden-Eijnden, 2023; Lipman et al., 2023; Liu et al., 2023). The training loss is

$$\min_\theta \frac{\Delta t}{n} \sum_{i=1}^{n} \sum_{j=1}^{m} \left\| v_\theta(X_{i,t_{j-1}}, t_{j-1}) - \frac{1}{\Delta t}(X_{i,t_j} - X_{i,t_{j-1}}) \right\|^2. \tag{24}$$

We optimize $\theta$ for $L_2$ steps using Adam. Since the particle update does not require access to the neural network $v_\theta$, and flow matching reduces the cost only when trajectories $X$ intersect, we apply the flow matching update less frequently.

In each loop, we compute on the batch $B$ of particles. The $L_1$ steps of particle update of trajectories $X^B$ can be computed by an innerloop on mini-batches within $B$, and same with the $L_2$ steps of velocity field update by flow matching on $X^B$.

**Computational cost** The main computational costs of Algorithm 1 come from the cost and particle update (line 4) and the flow matching update (line 5). The cost of the particle update scales as $O(BL_1)$, and that of the flow matching update scales as $O(BL_2)$, both proportional to the batch size and number of update steps. Importantly, the flow matching update is performed infrequently. In addition, compared to simulation-based methods, our approach offers a simpler and more efficient training process: the velocity network is updated via a decoupled, simulation-free flow matching step, which significantly reduces overall computational cost.

## D PROOFS

### D.1 PROOF OF PROPOSITION 2.1

The key for the proposition to hold is that, up to the change of variables $(\rho, m) = (\rho, \rho v)$, the continuity constraint becomes linear and the dynamic cost becomes convex. With this change of variable, it becomes straightforward that the fixed-point formulation (1) is equivalent to the first-order optimality condition of the variational problem (7).

*Proof.* Let $m = \rho v$ be the vector-valued Radon measure, and let $\frac{dm}{d\rho}$ denote its Radon–Nikodym derivative. Define

$$C_{(\rho,m)} := \left\{ (\tilde{\rho}, \tilde{m}) : \partial_t \tilde{\rho} + \nabla \cdot \tilde{m} = 0, \ \tilde{\rho}_0 = \mu, \ \tilde{\rho} \in C_\rho \right\}. \tag{25}$$

The constraint is linear in $(\tilde{\rho}, \tilde{m})$, hence the set $C_{(\rho,m)}$ is convex.

Define the functional

$$\mathcal{R}(\tilde{\rho}, \tilde{m}) = \begin{cases} \frac{1}{2} \int_0^1 \int \left\| \frac{d\tilde{m}_t}{d\tilde{\rho}_t}(x) \right\|^2 d\tilde{\rho}_t(x)\, dt, & \tilde{m}_t \ll \tilde{\rho}_t \ \forall t \in [0,1], \\ \infty, & \text{otherwise.} \end{cases} \tag{26}$$

Then $\mathcal{R}(\tilde{\rho}, \tilde{m})$ is convex in $(\tilde{\rho}, \tilde{m})$. Next, set

$$\mathcal{J}(\tilde{\rho}, \tilde{m}; \rho) := \mathcal{R}(\tilde{\rho}, \tilde{m}) + \int_0^1 \int F[\rho_t]\, d\tilde{\rho}_t(x)\, dt + \int G[\rho_1]\, d\tilde{\rho}_1(x). \tag{27}$$

The functional $\mathcal{J}(\tilde{\rho}, \tilde{m}; \rho)$ is convex in $(\tilde{\rho}, \tilde{m})$.

Since the objective is convex and the constraint is linear, $(\tilde{\rho}, \tilde{m})$ solves

$$\min_{(\tilde{\rho}, \tilde{m}) \in C_{(\rho,m)}} \mathcal{J}(\tilde{\rho}, \tilde{m}; \rho) \tag{28}$$

if and only if it satisfies the first-order optimality condition:

$$\begin{cases} -\partial_t \tilde{\phi} + \frac{1}{2}\|\nabla \tilde{\phi}\|^2 = F[\rho], & \tilde{\phi}_1 = G[\rho_1], \\ \tilde{m} = -\tilde{\rho}\nabla\tilde{\phi}, \\ \partial_t \tilde{\rho} + \nabla \cdot \tilde{m} = 0, & \tilde{\rho}_0 = \mu. \end{cases} \tag{29}$$

Here, the Hamilton-Jacobi-Bellman equation is understood to hold on the support of $\tilde{\rho}$, with equality replaced by inequality $-\partial_t \tilde{\phi} + \frac{1}{2}\|\nabla\tilde{\phi}\|^2 \leq F[\rho]$, $\tilde{\phi}_1 \leq G[\rho_1]$ outside the support. The Fokker–Planck equation is understood in the sense of distributions.

Therefore, $(\hat{\rho}, \hat{v})$ solves (1) if and only if it satisfies

$$
\begin{cases}
-\partial_t \hat{\phi} + \frac{1}{2}\|\nabla\hat{\phi}\|^2 = F[\hat{\rho}], & \hat{\phi}_1 = G[\hat{\rho}_1], \\
\hat{v} = -\nabla\hat{\phi}, \\
\partial_t \hat{\rho} + \nabla \cdot (\hat{\rho}\hat{v}) = 0, & \hat{\rho}_0 = \mu,
\end{cases}
\tag{30}
$$

Similarly, deriving the first-order optimality condition for (7) yields the same system (30). Hence, $(\hat{\rho}, \hat{v})$ solving (1) is necessary for solving (7), and becomes sufficient provided that $\mathcal{F}, \mathcal{G}$ are convex in $\rho$ as in Definition B.3. $\qquad\square$

### D.2 Proof of Lemma 4.1

We prove the Lemma in two steps. We first show that the distribution flow $\tilde{\rho}$ induced by $\tilde{X}$ is in $C_\rho$ and the interaction and terminal cost are unchanged with the change from Lagrangian coordinate $\tilde{X}$ to Eulerian coordinate $\tilde{\rho}$. As summarized in the following lemma.

**Lemma D.1.** *Let $\tilde{X} \in C_X$ and set $\tilde{\rho}_t := (\tilde{X}_t)_\# \mu$. Then $\tilde{\rho} \in C_\rho$, $\int F(x)\mathrm{d}\tilde{\rho}_t(x) = \mathbb{E}_{x\sim\mu}[F(\tilde{X}_t(x))]$ for any $F : \mathbb{R}^d \to \mathbb{R}^d$ and $\mathcal{F}(\tilde{\rho}_t) = \mathcal{F}((\tilde{X}_t)_\#\mu)$ for any $\mathcal{F} : \mathcal{P}_2(\mathbb{R}^d) \to \mathbb{R}$.*

*Proof.* The costs are unchanged as a direct result of the definition of pushforward.

We first prove that $\tilde{\rho}_t \in \mathcal{P}_2(\mathbb{R}^d)$. Fix $t \in [0, 1]$. For $\mu$-a.e. $x$, since $\tilde{X}(x, \cdot)$ is absolutely continuous, we have

$$
\tilde{X}(x, t) = x + \int_0^t \partial_s \tilde{X}(x, s)\mathrm{d}s.
\tag{31}
$$

Therefore

$$
\|\tilde{X}(x, t)\|^2 \leq 2\|x\|^2 + 2\left\|\int_0^t \partial_s \tilde{X}(x, s)\mathrm{d}s\right\|^2 \leq 2\|x\|^2 + 2\int_0^1 \|\partial_s \tilde{X}(x, s)\|^2 \mathrm{d}s,
\tag{32}
$$

where the second inequality is by the Cauchy-Schwarz inequality. Since $\mu \in \mathcal{P}_2(\mathbb{R}^d)$ and $\tilde{X}(x, \cdot) \in AC^2(0, 1; \mathbb{R}^d)$ for $\mu$-a.e. $x$, integrating over $\mu$ gives

$$
\int \|x\|^2 \mathrm{d}\rho_t(x) = \int \|\tilde{X}(x, t)\|^2 \mathrm{d}\mu(x) \leq 2\int \|x\|^2 \mathrm{d}\mu(x) + 2\int \int_0^1 \|\partial_s \tilde{X}(x, s)\|^2 \mathrm{d}s\mathrm{d}\mu(x) < \infty,
\tag{33}
$$

therefore $\tilde{\rho}_t \in \mathcal{P}_2(\mathbb{R}^d)$.

For any $0 \leq s < t \leq 1$, consider the coupling $\pi = (\tilde{X}_s, \tilde{X}_t)_\#\mu$ between $\tilde{\rho}_s$ and $\tilde{\rho}_t$. By definition, we have

$$
\begin{aligned}
W_2^2(\tilde{\rho}_s, \tilde{\rho}_t) &\leq \int_{\mathbb{R}^d \times \mathbb{R}^d} \|y - z\|^2 \mathrm{d}\pi(y, z) \\
&= \int_{\mathbb{R}^d} \|\tilde{X}_s(x) - \tilde{X}_t(x)\|^2 \mathrm{d}\mu(x) = \int_{\mathbb{R}^d} \left\|\int_s^t \partial_r X(x, r)\mathrm{d}r\right\|^2 \mathrm{d}\mu(x).
\end{aligned}
\tag{34}
$$

Let

$$
g(r) := \|\partial_r X(\cdot, r)\|_\mu = \left(\int_{\mathbb{R}^d} \|\partial_r X(x, r)\|^2 \mathrm{d}\mu(x)\right)^{\frac{1}{2}}.
\tag{35}
$$

Then $g \in L^2([0, 1])$ and

$$
W_2(\tilde{\rho}_s, \tilde{\rho}_t) \leq \left\|\int_s^t \partial_r X(\cdot, r)\mathrm{d}r\right\|_\mu \leq \int_s^t \|\partial_r X(\cdot, r)\|_\mu \,\mathrm{d}r = \int_s^t g(r)\mathrm{d}r.
\tag{36}
$$

Therefore $\tilde{\rho} \in C_\rho$. $\qquad\square$

We then construct the velocity field $\tilde{v}$ such that $(\tilde{\rho}, \tilde{v}) \in C_{(\rho, v)}$, show that $\tilde{v}$ is the unique solution to the FM, and the dynamic cost in $(\tilde{\rho}, \tilde{v})$ is bounded above by the dynamic cost in $\tilde{X}$.

*Proof of Lemma 4.1.* We construct $\tilde{v}$ and show that it solves the FM. Define the vector measure

$$\tilde{m}_t(B) := \int_{\{x: \tilde{X}(x,t) \in B\}} \partial_t \tilde{X}(x,t) \mathrm{d}\mu(x), \quad B \subset \mathbb{R}^d. \tag{37}$$

Then $\tilde{m}_t \ll \tilde{\rho}_t$ and by Radon-Nikodym theorem, there exist a Borel vector field $\tilde{v}_t$ defined $\rho_t$-a.e. such that $\tilde{m}_t(B) = \int_B \tilde{v}_t(y) \mathrm{d}\tilde{\rho}_t(y)$. This implies that,

$$\int \tilde{v}(\tilde{X}(x,t), t) - \partial_t \tilde{X}(x,t) \mathrm{d}\mu(x) \mathrm{d}t = 0, \tag{38}$$

Consider the FM problem

$$\min_v \int_0^1 \int \|v(\tilde{X}(x,t), t) - \partial_t \tilde{X}(x,t)\|^2 \mathrm{d}\mu(x) \mathrm{d}t. \tag{39}$$

(38) shows that $\tilde{v}$ satisfies the optimality condition of (39). Since the problem (39) is unconstrained with a convex objective, $\tilde{v}$ is therefore the solution to (39).

We then prove that $(\tilde{\rho}, \tilde{v})$ satisfies the continuity equation. By the chain rule,

$$\frac{\mathrm{d}}{\mathrm{d}t} \int \varphi(y) \mathrm{d}\tilde{\rho}_t(y) = \frac{\mathrm{d}}{\mathrm{d}t} \int \varphi(\tilde{X}(x,t)) \mathrm{d}\mu(x) = \int \nabla \varphi(\tilde{X}(x,t)) \cdot \partial_t \tilde{X}(x,t) \mathrm{d}\mu(x). \tag{40}$$

By the definition of $\tilde{v}$, for any $\varphi \in C_c^\infty(\mathbb{R}^d)$,

$$\int \nabla \varphi(y) \cdot \tilde{v}(y,t) \mathrm{d}\tilde{\rho}_t(y) = \int \nabla \varphi(\tilde{X}(x,t)) \cdot \partial_t \tilde{X}(x,t) \mathrm{d}\mu(x). \tag{41}$$

Combining (40) and (41) gives the desired weak formula of the continuity equation:

$$\frac{\mathrm{d}}{\mathrm{d}t} \int \varphi \mathrm{d}\tilde{\rho}_t = \int \nabla \varphi \cdot v_t \mathrm{d}\tilde{\rho}_t. \tag{42}$$

In the end, we show

$$\|\tilde{v}_t\|_{\tilde{\rho}_t}^2 \le \|\partial_t \tilde{X}(\cdot, t)\|_\mu^2. \tag{43}$$

This implies $\tilde{v}_t \in L^2(\tilde{\rho}_t)$ and therefore concludes $(\tilde{\rho}, \tilde{v}) \in C_{(\rho, v)}$. In addition, integrating (43) over time and combining with Lemma D.1 gives $\mathcal{J}(\tilde{\rho}, \tilde{v}; \rho) \le \mathcal{J}(\tilde{X}; \rho)$.

Define a linear funational $T$ on $L^2(\tilde{\rho}_t)$ by

$$T(\psi) := \int \psi(y) \cdot \mathrm{d}\tilde{m}_t(y) = \int \psi(\tilde{X}(x,t)) \cdot \partial_t \tilde{X}(x,t) \mathrm{d}\mu(x). \tag{44}$$

Then by Cauchu-Schwarz inequality in $L^2(\mu)$

$$|T(\psi)| \le \|\psi \circ \tilde{X}_t\|_\mu \|\partial_t \tilde{X}(\cdot, t)\|_\mu = \|\psi\|_{\tilde{\rho}_t} \|\partial_t \tilde{X}(\cdot, t)\|_\mu, \tag{45}$$

which implies that $T$ is a bounded linear functional on the Hilbert space $L^2(\tilde{\rho}_t)$ with operator norm

$$\|T\| \le \|\partial_t \tilde{X}(\cdot, t)\|_\mu. \tag{46}$$

By the Riesz representation theorem, there exists $v_t \in L^2(\tilde{\rho}_t)$ such that for every $\psi \in L^2(\tilde{\rho}_t)$,

$$T(\psi) = \int \psi(y) \cdot v_t(y) \mathrm{d}\tilde{\rho}_t(y). \tag{47}$$

Moreover,

$$\|v_t\|_{\tilde{\rho}_t}^2 = \|T\|^2 \le \|\partial_t \tilde{X}(\cdot, t)\|_\mu^2. \tag{48}$$

It suffices to show that $v_t = \tilde{v}_t$. Let $B \subset \mathbb{R}^d$ be a Borel set and $a \in \mathbb{R}^d$ be a fixed vector, then $L^2(\tilde{\rho}_t) \ni \psi(y) := a \mathbf{1}_B(y) = \begin{cases} a, & y \in B, \\ 0, & y \notin B. \end{cases}$ By definition (44), $T(\psi) = a \cdot \tilde{m}_t(B)$. On the other hand, by Riesz representation (47), $T(\psi) = a \cdot \int_B v_t(y) \mathrm{d}\tilde{\rho}_t(y)$. Therefore $a \cdot \tilde{m}_t(B) = a \cdot \int_B v_t(y) \mathrm{d}\tilde{\rho}_t(y)$ for any $a \in \mathbb{R}^d$ and any Borel $B \subset \mathbb{R}^d$. This implies that $v_t = \frac{\mathrm{d}\tilde{m}_t}{\mathrm{d}\tilde{\rho}_t}$ and $v_t = \tilde{v}_t$ except on a $\tilde{\rho}_t$-zero measure set, which concludes the proof. $\quad\square$

### D.3   PROOF OF LEMMA 4.2

This is a direct corollary of (Ambrosio et al., 2008, Prop. 8.1.8).

### D.4   PROOF OF THEOREM 4.3

*Proof.* We first prove (i). Since $(\rho^*, v^*)$ solves (1), $v^*$ is in $C_v$ and by Lemma 4.2, there exists $X^* \in C_X$ such that $(X_t^*)_{\#}\mu = \rho_t^*$ and $\mathcal{J}(X^*; \rho^*) = \mathcal{J}(\rho^*, v^*; \rho^*)$. For any other $\tilde{X} \in C_X$, by Lemma 4.1, there exist $(\tilde{\rho}, \tilde{v}) \in C_{(\rho,v)}$ such that $\mathcal{J}(\tilde{\rho}, \tilde{v}; \rho^*) \leq \mathcal{J}(\tilde{X}; \rho^*)$. Therefore, $X^*$ solves (11).

The same logic applies to MFC and concludes the first part of (ii).

For the second part of (ii), it suffices to prove $\mathcal{J}(\rho^*, v^*) = \mathcal{J}(\rho, v)$. By optimality of $(\rho, v)$, we have $\mathcal{J}(\rho^*, v^*) \geq \mathcal{J}(\rho, v)$. By Lemma (4.2), there is an $X \in C_X$ such that $\mathcal{J}(X) = \mathcal{J}(\rho, v)$. By Lemma (4.1) and optimality of $X^*$, we have $\mathcal{J}(\rho^*, v^*) \leq \mathcal{J}(X^*) \leq \mathcal{J}(X) = \mathcal{J}(\rho, v)$, which concludes the proof.  □

### D.5   PROOF OF LEMMA 4.4

*Proof.* The dynamic cost

$$\mathbb{E}_{x \sim \mu} \left[ \int_0^1 \frac{1}{2} \|\partial_t \tilde{X}(x, t)\|^2 dt \right] = \|\partial_t \tilde{X}\|^2_{\mu \otimes [0,1]} \tag{49}$$

is strictly convex in $\tilde{X}$, In addition, $F[\rho_t^{(k)}], G[\rho_1^{(k)}]$ are $L$-smooth and $\alpha_k < \frac{1}{L}$, therefore the objective of the update (13) is strongly convex. Therefore, the minimizer is unique if it exists.

Let $\{X^{(n)}\} \subset C_X$ be a minimizing sequence $\lim_{n \to \infty} \mathcal{J}(X^{(n)}; \rho^{(k)}) = \inf_{X \in C_X} \mathcal{J}(X; \rho^{(k)})$. Then by $L$-smoothness of $F[\rho_t^{(k)}]$ and $G[\rho_1^{(k)}]$, and $0 < \alpha_k < \frac{1}{L}$, we have that

$$\sup_n \left( \|\partial_t X^{(n)}\|_{\mu \otimes [0,1]} + \|X^{(n)}\|_{\mu \otimes [0,1]} + \|X_1^{(n)}\|_\mu \right) < \infty,$$

which implies $\{X^{(n)}\}$ is bounded in $H^1(0, 1; L^2(\mu))$. Therefore there exists a subsequence, still denoted as $X^{(n)}$, and $X^* \in H^1(0, 1; L^2(\mu))$ such that $X^{(n)} \to X^*$ weakly in $H^1(0, 1; L^2(\mu))$.

By weakly convergence in $H^1(0, 1; L^2(\mu))$ and weak lower semicontinuity of $L^2$ norm, $\|\partial_t X^*\|^2_{\mu \otimes [0,1]} \leq \liminf_{n \to \infty} \|\partial_t X^{(n)}\|^2_{\mu \otimes [0,1]}$.

By compact embedding theorem, there exists a subsequence, still denoted as $X^{(n)}$ such that $X^{(n)} \to X^*$ strongly in $C(0, 1; L^2(\mu))$ and therefore $X^{(n)} \to X^*$ strongly in $L^2(0, 1; L^2(\mu))$ and $X_1^{(n)} \to X_1^*$ strongly in $L^2(\mu)$. Combining with the kinetic term, we have $\mathcal{J}(X^*; \rho^{(k)}) \leq \liminf_{n \to \infty} \mathcal{J}(X^{(n)}; \rho^{(k)})$, which implies $X^*$ is the minimizer.

By the update rule, $X^{(k+\frac{1}{2})} = X^*$ and the decay property (15) holds.  □

### D.6   PROOF OF LEMMA 4.5

*Proof.* By Lemma 4.1, $\rho^{(k+1)} \in C_\rho$, $v_t^{(k+1)}$ is unique upto a $\rho_t$-zero measure set. Then by Lemma 4.1 and Lemma 4.2, we have

$$\mathcal{J}(X^{(k+1)}; \rho) = \mathcal{J}(\rho^{(k+1)}, v^{(k+1)}; \rho) \leq \mathcal{J}(X^{(k+\frac{1}{2})}; \rho), \tag{50}$$

which concludes the proof.  □

### D.7   PROOF OF THEOREM 4.6

*Proof.* Since $F, G$ are independent of $\rho$, combining Lemmas 4.4 and 4.5 with $\alpha_k = \alpha$ gives

$$\frac{1}{2\alpha} \left( \|X^{(k+\frac{1}{2})} - X^{(k)}\|^2_{\mu \otimes [0,1]} + \|X_1^{(k+\frac{1}{2})} - X_1^{(k)}\|^2_\mu \right) \leq \mathcal{J}(X^{(k)}) - \mathcal{J}(X^{(k+1)}) \tag{51}$$

Telescoping both sides and $\mathcal{J}(X) \geq \underline{\mathcal{J}}$ gives (16).

If $F, G$ are $\lambda$-convex, then for any $X, Y \in C_X$,

$$\lambda \left( \|X - Y\|^2_{\mu \otimes [0,1]} + \|X_1 - Y_1\|^2_\mu \right)$$

$$\leq \|\partial_t (X - Y)\|^2_{\mu \otimes [0,1]} + \int_0^1 \langle \nabla F(X_t) - \nabla F(Y_t), X_t - Y_t \rangle_\mu \mathrm{d}t + \langle \nabla G(X_1) - \nabla G(Y_1), X_1 - Y_1 \rangle_\mu$$

$$(52)$$

Take $X = X^{(k+1)}$ and $Y = X^*$. By the update rule and the optimality of $X^*$, we have

$$\lambda \left( \|X^{(k+1)} - X^*\|^2_{\mu \otimes [0,1]} + \|X_1^{(k+1)} - X_1^*\|^2_\mu \right)$$

$$\leq -\frac{1}{\alpha} \left( \langle X^{(k+1)} - X^{(k)}, X^{(k+1)} - X^* \rangle_{\mu \otimes [0,1]} + \langle X_1^{(k+1)} - X_1^{(k)}, X_1^{(k+1)} - X_1^* \rangle_\mu \right),$$

$$(53)$$

By algebra on the inner product

$$\langle a - b, a - c \rangle = \frac{1}{2} \left( \|a - b\|^2 + \|a - c\|^2 - \|b - c\|^2 \right), \tag{54}$$

Take $a = X^{(k+1)}, b = X^{(k)}, c = X^*$. Then, (53) gives

$$(2\lambda\alpha + 1)\left( \|X^{(k+1)} - X^*\|^2_{\mu \otimes [0,1]} + \|X_1^{(k+1)} - X_1^*\|^2_\mu \right)$$

$$+ \left( \|X^{(k+1)} - X^{(k)}\|^2_{\mu \otimes [0,1]} + \|X_1^{(k+1)} - X_1^{(k)}\|^2_\mu \right) \tag{55}$$

$$\leq \left( \|X^{(k)} - X^*\|^2_{\mu \otimes [0,1]} + \|X_1^{(k)} - X_1^*\|^2_\mu \right).$$

Therefore,

$$\left( \|X^{(k+1)} - X^*\|^2_{\mu \otimes [0,1]} + \|X_1^{(k+1)} - X_1^*\|^2_\mu \right) \leq \frac{1}{1 + 2\lambda\alpha} \left( \|X^{(k)} - X^*\|^2_{\mu \otimes [0,1]} + \|X_1^{(k)} - X_1^*\|^2_\mu \right),$$

$$(56)$$

and (17) holds. $\square$

# E  EXPERIMENT DETAILS

## E.1  2D TOY EXAMPLE

We use multilayer perceptrons (MLP) with six hidden layers of width 512 and Rectified Linear Unit (ReLU) activations to model the velocity field and the classifier. The time $t$ is concatenated with the input, and the concatenated input is fed into the velocity field.

## E.2  NON-POTENTIAL MFG

In this example, we consider an MFG on $\mathbb{R}^2$ with interaction cost defined by $F[\rho](x, t) = \lambda_F \int w(x, y) \mathrm{d}\rho(y)$ where $\lambda_F = 10$, $w(x, y) = \exp(a^\top (x - y))$ and $a = [0, 1]^\top$. Since the kernel $w$ is asymmetric, the game is therefore non-potential. Under this cost, a player at state $x$ incurs exponentially greater interaction cost from masses located ahead of it in the direction of $a$ than from masses behind. To reduce this cost, a player tends to move along the direction of $a$ if many players are ahead of it. The terminal cost is chosen to be independent of the population density: $G(x) = \lambda_G (a^\top x - c)^2$, where $\lambda_G = 1$, $a$ is the same as that in $F$ and $c = -1$ is fixed. To minimize this terminal cost, players must move toward the target hyperplane $a^\top x = c$. The initial distribution is chosen as a Gaussian with mean $[0, 1]^\top$ and a diagonal covariance matrix with entries $0.02$ and $0.1$. For this setting, evaluating the interaction and terminal costs and their gradients $\nabla F[\rho], \nabla G[\rho]$ only requires computing empirical averages of $e^{a^\top x}$ and does not involve neural network approximations.

To parametrize $v_\theta$, we use an MLP with 3 hidden layers of width [4, 8, 16]. The hyperparameters: $K = 100$, $B = 1000$, particle trained for $L_1 = 100$ steps, and cost updated every $L_0 = 2$ steps; flow matching trained for $L_2 = 100$ steps. Training performance is measured by the residual

$$\left( \frac{\Delta t}{n} \sum_{i=1}^n \sum_{j=1}^{m-1} \left\| -(D_{tt} X_i^{(k)})_{t_j} + \nabla F[\rho_{t_j}^{(k)}](X_{i,t_j}^{(k)}) \right\|^2 + \frac{1}{n} \sum_{i=1}^n \left\| (D_t X_i^{(k)})_{t_m} + \nabla G[\rho_{t_m}^{(k)}](X_{i,t_m}^{(k)}) \right\|^2 \right)^{\frac{1}{2}},$$

$$(57)$$

which is a sample-based approximation of the $\mu \otimes [0, 1]$ norm of the first-order variation of $\mathcal{J}(X; \rho^{(k)})$ at $X = X^{(k)}$. The residual reflects how much the necessary condition of the MFG is violated; that is, a small residual indicates proximity to a stationary point. Figure 3 (top left) shows that the algorithm converges to a state with a residual of $10^{-1}$.

We visualized the learned flow on test trajectories in the top right plot of Figure 3 and visualize the density evolution, estimated using Gaussian kernel density estimation in the bottom plot. All players move toward the line $x_2 = -1$ to reduce the terminal cost. For players with initially larger $x_2$ values, the interaction cost dominates early in the trajectory since there is more mass ahead of them along $a$. To mitigate this, they move quickly in the direction of $a$. Players starting at smaller $x_2$ values also tend to move in the direction of $a$ to avoid falling behind when other players catch up. However, because their initial interaction cost is lower, they balance interaction and dynamic costs, leading to slower initial movement. Although the initial distribution has a larger variance along the $x_2$ direction, players' desire to reduce interaction cost causes them to compress along $x_2$ and spread out along $x_1$ after $t = 0.25$, making $x - y$ nearly orthogonal to $a$. Under this population profile, the interaction cost motivates agents to match the pace of the group while moving toward the target terminal location $x_2 = -1$. The behavior is consistent with the physical interpretation of the costs and equilibrium.

### E.3 Image-to-image translation

We evaluate the quality of translated images by using Fréchet inception distance (FID) (Heusel et al., 2017) and compare with different baselines: flow matching methods including Optimal Transport Conditional Flow Matching (OT-CFM) (Tong et al., 2023), Stochastic Interpolants (SI) (Albergo & Vanden-Eijnden, 2023), Rectified Flow (Liu et al., 2023), Schrödinger Bridge Conditional Flow Matching (SB-CFM) (Tong et al., 2023); two representative generative adversarial networks (GANs), Disco GAN (Kim et al., 2017) and Cycle GAN (Zhu et al., 2017); Neural Optimal Transport (NOT) (Korotin et al., 2023); and Q-flow (Xu et al., 2025).

For image data, we first train a deep variational autoencoder (VAE) to compress images into a latent space. The deep VAE is trained on the same setup following (Rombach et al., 2022) in a latent space of $\mathbb{R}^{12 \times 8 \times 8}$, enabling the trajectory to be learned in the latent space across all methods. We use a constant learning rate of 4.5e-06 and a batch size of 32 with gradient accumulation over 8 batches for training VAE on both datasets. For the handbags–shoes dataset, a single VAE is trained for 100 epochs, using 90% of the images for training and the remaining 10% for testing. For the CelebA dataset, we train a single VAE for 100 epochs, adopting the standard training, validation, and test splits from `torchvision.datasets.CelebA`. Table 3 reports the FID between reconstructed images from the training set and ground-truth images from the test set (handbags, shoes, CelebA male, and CelebA female), presenting the quality of the trained VAE.

We use a convolutional U-Net architecture (Ronneberger et al., 2015) for the classifier. The Swish activation function is used throughout the network, except for the final fully connected layer, where ReLU is used. For the velocity field, we use a stack of convolutional layers with ReLU activations. The time $t$ is concatenated with the input along the channel dimension, and the concatenated input is fed into each convolutional layer. The same velocity field architecture is used for our method and the reproduced baselines reported. The detailed architectures of the classifier and velocity field are provided in Table 4 and Table 5, respectively.

In our method, we initialize the velocity field using FM with a linear interpolant function. The hyperparameters: $K = 50$, $B =$10k for handbag-to-shoe and 15k for CelebA, particle trained for $L_1 = 1000$ steps for handbag-to-shoe and 300 steps for CelebA, and cost updated every $L_0 = 10$ steps; velocity field trained by flow matching for $L_2 = 1000$ steps for handbag-to-shoe and 300 steps for CelebA. For both the handbags-to-shoes and CelebA datasets, the reproduced baselines (OT-CFM, Stochastic Interpolants, Rectified Flow, SB-CFM) are trained for 400K steps with a batch size of 512.

Table 3: FID between reconstructed images from the training set and ground-truth images from the held-out sets on the images of Handbags, shoes, CelebA male, and CelebA female.

| Handbag | Shoes | Male | Female |
|---------|-------|------|--------|
| 5.07 | 6.09 | 7.09 | 5.06 |

Table 4: Architecture details of the classifier.

| | Layer | Component |
|---|---|---|
| **Encoding** | Conv 1 | $3 \times 3$ `Conv2d`, channels=256, kernel size=3, stride=1, bias=True, padding=1 |
| | Group Normalization 1 | num groups=4, channels=256 |
| | Conv 2 | $3 \times 3$ `Conv2d`, channels=512, kernel size=3, stride=1, bias=True, padding=1 |
| | Group Normalization 2 | num groups=32, channels=512 |
| | Conv 3 | $3 \times 3$ `Conv2d`, channels=512, kernel size=3, stride=2, bias=True, padding=1 |
| | Group Normalization 3 | num groups=32, channels=512 |
| | Conv 4 | $3 \times 3$ `Conv2d`, channels=1024, kernel size=3, stride=1, bias=True, padding=1 |
| | Group Normalization 4 | num groups=32, channels=1024 |
| | Conv 5 | $3 \times 3$ `Conv2d`, channels=1024, kernel size=3, stride=1, bias=True, padding=1 |
| | Group Normalization 5 | num groups=32, channels=1024 |
| **Decoding** | Conv Transpose 5 | $3 \times 3$ `ConvTranspose2d`, channels=1024, kernel size=3, stride=1, bias=True, padding=1 |
| | Group Normalization 5 | num groups=32, channels=1024 |
| | Conv Transpose 4 | $3 \times 3$ `ConvTranspose2d`, channels=1024, kernel size=3, stride=1, bias=True, padding=1 |
| | Group Normalization 4 | num groups=32, channels=1024 |
| | Conv Transpose 3 | $4 \times 4$ `ConvTranspose2d`, channels=512, kernel size=4, stride=2, bias=True, padding=1 |
| | Group Normalization 3 | num groups=32, channels=256 |
| | Conv Transpose 2 | $3 \times 3$ `ConvTranspose2d`, channels=512, kernel size=3, stride=1, bias=True, padding=1 |
| | Group Normalization 2 | num groups=32, channels=512 |
| | Conv Transpose 1 | $3 \times 3$ `ConvTranspose2d`, channels=256, kernel size=3, stride=1, bias=True, padding=1 |

Table 5: Architecture details of the velocity field.

| | Layer | Component |
|---|---|---|
| **Encoding** | Conv 1 | $3 \times 3$ Conv2d, channels=64, stride 1, bias=True, padding=1 |
| | Conv 2 | $3 \times 3$ Conv2d, channels=256, stride 1, bias=True, padding=1 |
| | Conv 3 | $3 \times 3$ Conv2d, channels=512, stride 2, bias=True, padding=1 |
| | Conv 4 | $3 \times 3$ Conv2d, channels=512, stride 1, bias=True, padding=1 |
| | Conv 5 | $3 \times 3$ Conv2d, channels=1024, stride 1, bias=True, padding=1 |
| **Decoding** | Conv Transpose 5 | $3 \times 3$ ConvTranspose2d, channels=1024, stride 1, bias=True, padding=1 |
| | Conv Transpose 4 | $3 \times 3$ ConvTranspose2d, channels=512, stride 1, bias=True, padding=1 |
| | Conv Transpose 3 | $4 \times 4$ ConvTranspose2d, channels=512, stride 2, bias=True, padding=1 |
| | Conv Transpose 2 | $3 \times 3$ ConvTranspose2d, channels=256, stride 1, bias=True, padding=1 |
| | Conv Transpose 1 | $3 \times 3$ ConvTranspose2d, channels=64, stride 1, bias=True, padding=1 |

