# OpenReview forum: "High-dimensional Mean-Field Games by Particle-based Flow Matching"
_ICLR.cc/2026/Conference — ICLR 2026 Poster_

### Official Review · Reviewer_HG67 · 2025-10-27

**Soundness:** 3
**Presentation:** 3
**Contribution:** 2
**Rating:** 4
**Confidence:** 2

**Summary:**

The paper is concerned with solving potentially high-dimensional Mean-Field Games (MFGs), which are challenging due to their optimal control fixed-point structure. Existing solvers often rely on mesh-based discretizations, and as a result become computationally infeasible when applied to high-dimensional problems. To overcome this, the authors propose a particle-based deep flow matching method along with a trust-region fixed-point scheme for efficiently solving high-dimensional MFGs.

**Strengths:**

1. The paper is well written and easy to follow.
2. The claims are justified with proofs whenever necessary.

**Weaknesses:**

1. Since the focus is on solving high-dimensional problems, out of the three example cases, it seems only the image-to-image translation results are noteworthy.

2. Line 473: "We can observe that our method produces smooth and coherent translations, particularly in terms of color consistency and reduction of visual artifacts." The authors only present the results obtained using their method, and this claim is rather relative than absolute. Hence, results using alternative methods are perhaps needed to support this claim/observation.

**Questions:**

Given that there exist deep learning implementations for solving MFGs, would it be possible to compare against such methods (I was able to find some below)? I think this result would greatly strengthen the paper if indeed the proposed method converges orders of magnitude faster as hinted in the paper.


[1] Chen, X., et al., *Physics-Informed Neural Operator for Coupled Forward-Backward Partial Differential Equations*.

[2] Chen, X., et al., *A Hybrid Framework of Reinforcement Learning and Physics-Informed Deep Learning for Spatiotemporal Mean Field Games*.

---

> ### Author Response · Authors · 2025-12-03
> **Authors' Rebuttal**
>
> ### W1. High-dimensional examples
> Thank you for pointing this out.
>
> We believe that the image-to-image translation task is the most representative of high-dimensional settings (as widely adopted in related studies [A, B, C]) and illustrates the practical relevance of our method. As one of the few numerical MFG studies to include such an experiment, we believe it highlights the scalability and potential of our approach.
>
> Due to time constraints, we were only able to include image-to-image translation task as our high-dimensional example in this submission. However, in the revised version, we have expanded the set of comparison methods based on the prior works to provide a more comprehensive evaluation.
>
> We fully acknowledge the importance of broader empirical validation and plan to explore additional high-dimensional applications in future work.
>
> - [A] Computing high-dimensional optimal transport by flow neural networks, Xu et al., AISTATS 2025
>
> - [B] Diffusion Schrödinger Bridge Matching, Shi et al., NeurIPS 2023
>
> - [C] Improving and generalizing flow-based generative models with minibatch optimal transport, Tong et al., TMLR 2024
>
>
> ### W2. Claim of numerical results
> Thank you for your helpful comment.
>
> We would like to clarify that the purpose of the visual examples is to assist readers in qualitatively assessing the output of our method. For quantitative comparison, we refer to the FID (Fréchet Inception Distance) scores presented in Table 1, where we compare our method with flow-based baselines. The FID is a widely used metric for assessing the quality of images generated by generative models. As shown in Table 1, our method generally outperforms the baselines in terms of FID score, supporting our claims regarding visual quality.
>
>
> ### Q1. Related works
> Thank you for pointing out these relevant works.
>
> We would like to clarify that our paper does not claim faster convergence rates compared to existing methods. Instead, we provide a theoretical convergence analysis for a subclass of MFGs, optimal control problems, and emphasize that our approach is simulation-free, which generally results in lower computational cost than simulation-based methods.
> By “simulation-free,” we refer to training methods where the forward pass does not require solving ODEs, PDEs, or SDEs. In contrast, simulation-based approaches often incur higher computational cost during backpropagation due to the need to differentiate through the solver used in the forward pass.
>
> Regarding the suggested references:
> - [1] focuses on learning a solution operator that can generalize across a family of MFG problems (e.g., with varying initial conditions but fixed cost structure). This is a different objective from our work, which aims to solve a single MFG instance efficiently and scalably.
> - [2] relies on mesh-based spatial discretization, which makes it less suitable for high-dimensional problems, which is the primary focus of our method.
>
> We believe our method complements existing work by targeting scalable training in high-dimensional settings, and we appreciate the reviewer’s suggestion to clarify this distinction in the revised manuscript.

---

### Official Review · Reviewer_pZrr · 2025-10-31

**Soundness:** 3
**Presentation:** 4
**Contribution:** 4
**Rating:** 6
**Confidence:** 3

**Summary:**

This paper proposes method for solving deterministic mean field games by combining a particle-based approach with a flow network by alternatively optimizing particle trajectories to minimize cost and the flow network to match the optimized particles. The proposed numerical scheme is shown to converge sublinearly for potential MFGs and linearly for optimal control problems under suitable assumptions. Numerical experiments on a non-potential MFG toy example confirm the convergence of the proposed scheme, while an image-to-image translation task on a learned latent space shows the proposed method produces smooth transitions and yields lower FID compared to baseline methods.

**Strengths:**

- To the best of my knowledge, the proposed method for solving deterministic mean field games is novel in the literature
- The writing is clear and easy to understand
- The empirical results help to demonstrate the effectiveness of the proposed method

**Weaknesses:**

- While the authors claim to solve (general) mean field games, it seems like the work only tackles **deterministic** mean field games, i.e., where the dynamics of each agent do not have any stochastic terms (e.g., equation in Line 170 doesn’t any diffusion related terms, and the evolution of the particles in (6) is an ODE and not a SDE). This could be somewhat misleading to readers, especially as this is not stated in either the title, abstract, or in the main body of the paper.

**Questions:**

- In relation to the above comment on the deterministic dynamics, it would be helpful to the readers for the author to cite and briefly discuss these additional works that do tackle this subclass, (e.g., [A, B]) and potentially compare against some of these methods (though not required), as well as ones that don’t make this assumption (e.g., [C, D, E, F]).
- Minor: Typo with “following” in line 469

[A] Gomes, Diogo, Julian Gutierrez, and Mathieu Lauriere. "Machine learning architectures for price formation models." *Applied Mathematics & Optimization* 88.1 (2023): 23.

[B] Assouli, Mouhcine, et al. "Initialization-driven neural generation and training for high-dimensional optimal control and first-order mean field games." *arXiv preprint arXiv:2507.15126* (2025).

[C] Lin, Alex Tong, et al. "Alternating the population and control neural networks to solve high-dimensional stochastic mean-field games." *Proceedings of the National Academy of Sciences* 118.31 (2021)

[D] Liu, Guan-Horng, et al. "Deep generalized schrödinger bridge." *Advances in Neural Information Processing Systems* 35 (2022)
[E] Chen, Yongxin. "Density control of interacting agent systems." *IEEE Transactions on Automatic Control* 69.1 (2023)

[F] Liu, Guan-Horng, et al. "Generalized Schr\" odinger Bridge Matching." *arXiv preprint arXiv:2310.02233* (2023).

---

> ### Author Response · Authors · 2025-12-03
> **Authors' Rebuttal**
>
> ### W1: Confusion due to lack of clarity about the deterministic MFG setting
> Thank you for pointing this out.
>
> We apologize for the oversight. We have clarified in both the abstract and at the end of the second paragraph of the introduction that our focus is on stochastic (first-order) MFGs.
>
> We would also like to emphasize that deterministic MFGs are not necessarily easier than stochastic (second-order) MFGs, where the dynamics include noise. Specifically:
> - Theoretical complexity: Solutions to deterministic MFGs are typically defined in a weaker sense, and their existence often requires stronger assumptions compared to stochastic MFGs.
> - Numerical challenges of PDE based solvers: In low-dimensional settings, mesh-based methods are commonly used to represent the density $\rho$ and value function $\phi$. When the dynamics are deterministic, the corresponding PDE formulation involves first-order Hamilton–Jacobi–Bellman (HJB) equations and Fokker–Planck (FP) equations, which can suffer from instability due to characteristic intersection. To address this, numerical schemes often introduce artificial diffusion (noise) to make the equations second-order, as this improves stability.
> - Numerical challenges of optimization-based solvers: In low-dimensional settings, these avoid directly solving the PDEs but can struggle with conditioning when the density becomes small. In first-order MFGs, the absence of a global lower bound on $\rho$ may lead to poor conditioning, requiring either very small step sizes (proportional to the inverse of the condition number) or resulting in unstable updates.
>
> Our method addresses the challenges of both mesh-based solvers:
> - It handles characteristic intersections by allowing trajectories to intersect and using flow matching and resampling to regularize the resulting trajectories.
> - It avoids small-density instability by relying on sample-based updates, where trajectories are naturally concentrated in high-probability regions.
>
> We acknowledge that our method is currently focus on deterministic MFGs. However, we believe it has strong potential for extension to second-order MFGs, especially given recent advances in machine learning on simulating and matching stochastic differential equations.
>
> ### Q1. Improve “Related Works” section
> Thank you for the suggestion.
>
> We have revised the “Related Works” section and added citations to [A, B, C] that are more related to our work.
>
> To clarify how our work differs from existing approaches, we highlight three key objects in MFGs: the population distribution $\rho$, the control $v$, and the value function $\phi$ (which arises as a Lagrange multiplier in the optimal control formulation of problem (1).
>
> Our method primarily updates the population $\rho$ through particle-based optimization; the control $v$ is obtained via flow matching; and the value function $\phi$ can also be recovered from particle trajectories.
> - [A] formulates MFGs as a saddle-point problem and solves it via a descent-ascent method. Both the control $v$ and value function $\phi$ are parameterized with neural networks and updated at each iteration. In contrast, we treat MFGs as a fixed-point problem and focus on updating particles to represent $\rho$, with $v$ recovered via flow matching.
> - [B] simulates fictitious play by jointly parameterizing $\rho$ and $\phi$, training them via penalization of the FP and HJB, respectively. In contrast, our method avoids explicit PDE constraints and relies on a sample-based formulation.
> - [C] also formulates the MFG as a saddle-point problem and updates parameterized $\rho$ and $\phi$ using penalties on FP and HJB residuals, similar to [B].
> - [D] considers a mean-field planning problem with hard terminal constraints, similar to optimal transport. Their method alternates between fitting the source and target distributions, which leverages this special structure. Our work considers general terminal cost function, not necessarily the indicator function of a  distribution.
> - [E] proposes an iterative algorithm that requires solving a discrete static OT problem at each step, which may not scale well to high-dimensional settings. Our work can scale to high dimensions and work on image-to-image translation tasks.
> - [F] studies stochastic optimal control rather than MFGs.
>
> ### Q2. Typo
> Thank you for pointing this out. We revised the writing and corrected typos.

---

### Official Review · Reviewer_669D · 2025-11-01

**Soundness:** 3
**Presentation:** 3
**Contribution:** 2
**Rating:** 4
**Confidence:** 3

**Summary:**

The paper proposes solving mean field games (MFGs) using a particle-based flow matching method. For a certain type of MFG, the proposed method replaces the backward best-response calculation and forward population simulation in a typical fixed-point iteration method with first-order gradient updates and particle-based flow matching. Theoretical convergence guarantees are provided for potential MFGs and the optimal control setting.

**Strengths:**

- The paper tackles the important challenge faced by fixed-point iteration methods for solving MFGs.
- The proposed method is novel and reasonable.
- The paper is mathematically rigorous and provides theoretical convergence analysis for the proposed method for certain classes of MFGs.

**Weaknesses:**

### Optimal Transport or Mean Field Game?

The model considered in this work is heavily inspired by optimal transport models but does not have much MFG flavor.

First, the cost function has a very special structure: the running cost is separable into a control-related cost and a population-related cost. Furthermore, the control-related cost term is a simple quadratic function. These are two significant simplifications of the cost function that are not satisfied by general MFGs.

Second, a very special dynamics model, an Euclidean spatial state space with deterministic velocity control, is considered. This imposes strong assumptions, including continuity and determinism in the state evolution and that agents have full knowledge of the dynamics. General games do not satisfy these assumptions.

Third, because of the simple dynamics model considered, agents' transitions are independent of the population distribution. This significantly reduces agents' coupling and thus weakens the model's connection to the general dynamic MFG framework, where both agents' rewards and transitions depend on the population distribution.

These model considerations and simplifications make sense from the perspective of optimal transport, but they substantially limit the applicability of the proposed method to general MFGs.

### The Claim of "Simulation-Free" and Its Benefits

It is mentioned multiple times that the proposed method is "simulation-free", meaning that it does not simulate a PDE/ODE system. However, the trajectories of the particles still need to be simulated according to ODE (6). I do not see the fundamental difference between these two types of "simulation".

Furthermore, it is claimed that other deep flow methods that simulate the population struggle in high-dimensional settings. However, it is not clear to me what the benefits of a particle-based flow matching method are in this regard. Specifically, the high-dimensional nature also requires an (exponentially) large number of particle trajectories $\{ X_{i,t_{j}}^{(k)} \}_{i,j}^{n,m}$ to approximate the true trajectory field $X^{(k)}$.

### Related Work

Recognizing the challenges faced by fixed-point iteration methods, several recent works \[1,2,3\] also propose simple methods that eliminate the forward-backward (best-response and population simulation) structure and demonstrate better convergence properties. These works are closely related to the discussions in the paper:

- They are generalized Frank–Wolfe algorithms that update parameters using first-order gradient information. Thus, these methods do not need to calculate the exact best response or the induced population at each iteration. They share the same motivation that "when the objective changes at each step, moving toward the best response of the current step eventually leads to an MFNE" (Lines 78–79) and that "convergence to a fixed point can still be achieved as long as each update yields improvement with respect to the current objective" (Lines 293–294).
- They are truly simulation-free methods, as the population distribution is learned in a model-free manner and can be estimated from a single trajectory of a single representative agent.
- \[3\] addresses continuous state–action spaces by incorporating function approximation.

\[1\]: Angiuli, A., Fouque, J.-P., Laurière, M., 2022. Unified reinforcement Q-learning for mean field game and control problems. Mathematics of Control, Signals, and Systems 34, 217–271.
\[2\]: Zeng, S., Bhatt, S., Koppel, A., Ganesh, S., 2025. Learning in herding mean field games: Single-loop algorithm with finite-time convergence analysis, in: International Conference on Artificial Intelligence and Statistics. PMLR.
\[3\]: Zhang, C., Chen, X., Di, X., 2025. Stochastic semi-gradient descent for learning mean field games with population-aware function approximation, in: International Conference on Learning Representations. PMLR.

**Questions:**

- Why do you call the update rule (8) a "trust-region strategy"? Where is the trust-region radius, its update rule, and the rejection rule? Isn’t it a soft regularization method? Do you mean that you want to constrain the next iteration to be near the current iteration?
- It is stated that the proposed method circumvents the "main challenge in solving MFGs … enforcing the continuity equation constraint." Is it because it learns a velocity field that matches the trajectories, so it automatically satisfies the continuity equation? Am I correct in understanding that other works enforce the continuity equation by simulating a population consistent with the control, while you learn a control consistent with the population trajectories?
- How are $n_{1}$ and $n_{2}$ related to $n$ in Algorithm 1?
- What is the non-potential MFG considered in Section 4.2? Is it purely illustrative or a model for real applications?

Please also see the questions in the Weaknesses section.

---

> ### Author Response · Authors · 2025-12-03
> **Authors' Rebuttal 1**
>
> ### W1. Strong assumptions
> Thank you for the comment.
>
> We respectfully disagree with the claim that our assumptions oversimplify the setting in general MFGs. Our setting aligns with the standard formulation of continuous-state MFGs that was originally proposed in [A] and considered in many computational works, e.g. [B, C].
> - Separable cost and quadratic dynamic cost
>   - A core feature of MFGs is the interaction between individuals and the population. Our model retains population-dependent cost terms.
>   - We adopt a separable Hamiltonian to decouple dynamic and interaction costs which is a common assumption in the literature. Importantly, our method does not critically rely on this separability and can be extended to more general (non-separable) settings.
>   - Regarding the control-related cost term, we use a quadratic function primarily because it is prevalent in machine learning applications. However, our approach is applicable to more general dynamic cost $L(v)=\|v\|_p^p$, and our analysis can be adapted accordingly.
> - Deterministic dynamics in Euclidean space.
>   - Modeling the state space as Euclidean does not simplify the problem, particularly in high dimensions, where enforcing the continuity equation becomes challenging. A key contribution of our work is the use of flow matching to overcome this difficulty. We provide theoretical justification showing that flow matching preserves the time-marginal distributions of particles while also reducing dynamic cost.
>   - While we consider deterministic dynamics, we emphasize that deterministic (first-order) MFGs are not necessarily easier than stochastic (second-order) ones. Specifically:
>     - Theoretical complexity: Solutions to deterministic MFGs are typically defined in a weaker sense, and their existence often requires stronger assumptions compared to stochastic MFGs.
>     - Numerical challenges of PDE based solvers: In low-dimensional settings, mesh-based methods are commonly used to represent the density $\rho$ and value function $\phi$. When the dynamics are deterministic, the corresponding PDE formulation involves first-order Hamilton–Jacobi–Bellman (HJB) equations and Fokker–Planck (FP) equations, which can suffer from instability due to characteristic intersection. To address this, numerical schemes often introduce artificial diffusion (noise) to make the equations second-order, as this improves stability.
>     - Numerical challenges of optimization-based solvers: In low-dimensional settings, these avoid directly solving the PDEs but can struggle with conditioning when the density becomes small. In first-order MFGs, the absence of a global lower bound on $\rho$ may lead to poor conditioning, requiring either very small step sizes (proportional to the inverse of the condition number) or resulting in unstable updates.
>
>     Our method addresses the challenges of both mesh-based solvers:
>     - It handles characteristic intersections by allowing trajectories to intersect and using flow matching and resampling to regularize the resulting trajectories.
>     - It avoids small-density instability by relying on sample-based updates, where trajectories are naturally concentrated in high-probability regions.
>
>     We acknowledge that our method is currently focus on deterministic MFGs. However, we believe it has strong potential for extension to second-order MFGs, especially given recent advances in machine learning on simulating and matching stochastic differential equations.
>
> - Finally, regarding the agent dynamics: because the cost function includes an interaction term, each agent’s reward and evolution is inherently dependent on the population distribution. In continuous-state settings, it is standard to model stochasticity (as in discrete-state transitions) as state- and time-independent noise. Our work focuses on the zero-noise limit, i.e., the deterministic setting. The associated challenges and how we address them are detailed above.
>
> [A] Lasry, Jean-Michel, and Pierre-Louis Lions. "Mean field games." Japanese journal of mathematics 2, no. 1 (2007): 229-260.
>
> [B] Ruthotto, Lars, Stanley J. Osher, Wuchen Li, Levon Nurbekyan, and Samy Wu Fung. "A machine learning framework for solving high-dimensional mean field game and mean field control problems." Proceedings of the National Academy of Sciences 117, no. 17 (2020): 9183-9193.
>
> [C] Gomes, Diogo, Julian Gutierrez, and Mathieu Lauriere. "Machine learning architectures for price formation models." Applied Mathematics & Optimization 88, no. 1 (2023): 23.

---

> ### Author Response · Authors · 2025-12-03
> **Authors' Rebuttal 2**
>
> ### W2. “Simulation-free” and its benefits
> Thank you for the question.
>
> We would like to clarify what we mean by “simulation-free.” This refers to training methods where the forward pass does not require solving ODEs, PDEs, or SDEs. In our method, the neural network $v_{\theta}$ is trained via regression, and although it represents a velocity field, its forward evaluation does not involve solving any differential equations. Hence, the method is simulation-free.
> In contrast, simulation-based methods often require solving ODEs, PDEs, or SDEs during the forward pass, and backpropagating through these solvers can be computationally expensive, especially in high-dimensional settings.
>
> We agree with the reviewer that if the desired distribution is supported on a $d$-dimensional manifold, then to obtain an $\varepsilon$-accurate approximation, the number of samples required to capture the dynamics can scale as $O(\exp(d/\varepsilon))$. This is the intrinsic difficulty that no method can avoid in such a setting.
>
> However, in machine learning, it is commonly assumed under the manifold hypothesis that many high-dimensional datasets actually concentrate near low-dimensional manifolds. Sample-based methods, like ours, naturally adapt to such structures through the sampling process itself.
>
>
>
>
> ### W3. Related work
> Thank you for suggesting these relevant references.
>
> We would like to highlight that our focus is on solving MFGs in high-dimensional continuous state-action spaces, whereas [1] and [2] primarily address MFGs in finite state-action settings. In particular, [1] handles continuous spaces by discretizing the domain, which becomes impractical in high-dimensional problems due to the curse of dimensionality.
> We have updated the “Related Work” section to clarify this distinction.
>
> While we acknowledge that [3] considers continuous state-action spaces using function approximation, their approach relies on representing the solution as a linear combination of predefined basis functions. In high dimensions, reducing approximation error often requires a large number of basis functions, which significantly increases computational complexity. Moreover, effectively exploiting low-dimensional manifold structures in such cases typically demands careful and problem-specific basis design.
> In contrast, our method is fully sample-based. The sampled trajectories naturally concentrate on the support of the distribution and can automatically adapt to potential low-dimensional structures in the data and problem, without requiring explicit basis construction.

---

> ### Author Response · Authors · 2025-12-03
> **Authors' Rebuttal 3**
>
> ### Q1. Improper usage of “trust-region”.
> Thank you for the question.
>
> By “trust region,” we originally referred to restricting the best response search to a local neighborhood around the current state. However, we acknowledge that this terminology may be confusing, particularly due to its different meaning in statistical contexts.
> To avoid ambiguity, we have replaced “trust region” with “proximal,” which more accurately reflects the structure of our method. The radius of the "trust region" is the step size of our proximal update, denoted as $\alpha_{\ell}$ in equations (12).
>
>
> ### Q2. Difficulty of continuity equation
>
> Thank you for the question.
>
> Your understanding is absolutely correct. This also highlights the key distinction between our method and simulation-based approaches and why our method is simulation-free.
>
> We would also like to emphasize that we provide a theoretical guarantee in Lemma 4.1, showing that the flow-matched velocity field is consistent with the probability distribution flow induced by the particle trajectories. This result is crucial for justifying:
> - the Lagrangian formulation (equation (9)),
> - the descent property of flow-matching update (equations (13) and (11)), and
> - the convergence of the proposed algorithm.
>
>
>
> ### Q3. Explanation of Hyperparameters
> Thank you for the question.
>
> In each iteration, we first sample $n$ particles and compute their corresponding trajectories. Then both the particle update and the flow-matching update are performed using stochastic gradient-type methods. Specifically, $n_1$ denotes the batch size of trajectories used for the particle update, and $n_2$ denotes the batch size of trajectories used for training the flow-matching network.
> To improve clarity, we have added a summary of these notations in Appendix A.
>
>
> ### Q4. Non-potential MFGs
> Thank you for the question.
>
> Our original goal in including the non-potential MFG example was to illustrate the effectiveness of our method, and it served primarily as a conceptual demonstration. We agree that the original model was abstract and lacked a clear physical interpretation.
> In the revised manuscript, we have updated the non-potential MFG example in Section 5.2 (formerly Section 4.2) with a new example that offers a more meaningful physical interpretation.

---

### Meta-Review · Area_Chair_wkh4 · 2026-01-06

**Summary:**

This paper proposes a particle-based flow matching approach for solving deterministic mean field games, with the goal of avoiding the forward–backward simulation structure common in fixed-point methods. Reviewers generally agree that the method is technically sound, novel, and clearly presented, and that it is supported by rigorous theoretical analysis for relevant subclasses of MFGs. The main concerns center on the scope and positioning of the contribution. In particular, the formulation relies on structural assumptions on the dynamics and cost that limit direct applicability to more general MFG settings and place the method closer to deterministic control and optimal transport formulations. In addition, empirical validation in high-dimensional settings remains limited, with image-to-image translation serving as the primary large-scale example. Overall, reviewers view the approach as promising and well executed, with discussion mainly focused on whether the current scope and empirical coverage are sufficient to meet the acceptance bar.

**Reviewer Concerns:**

The rebuttal addresses many of the reviewers’ concerns in a satisfactory manner. The authors clearly clarify that the method is designed for deterministic MFGs and revise the manuscript accordingly, resolving potential confusion about the problem setting. They also strengthen the related work section, better positioning the method relative to saddle-point, fictitious-play, and PDE-penalization approaches. Additional explanations regarding the handling of the continuity equation, algorithmic hyperparameters, and the illustrative nature of the non-potential MFG example further improve clarity.

Some concerns remain partially outstanding. Reviewer 669D’s observation that the modeling assumptions bring the formulation closer to optimal transport than to fully general dynamic MFGs is largely conceptual and reflects differing perspectives on scope rather than a clear technical flaw. Reviewer HG67’s request for broader high-dimensional empirical validation beyond image translation is acknowledged, though not fully addressed within the current submission.

**Reviewer Scores:**

Reviewer 669D may increase their score from 4 to 6, as the authors clarify most of the conceptual concerns raised in the initial review.

Reviewer pZrr is likely to retain their original score of 6. The initial review was generally positive, and while the rebuttal clarifies the focus on deterministic MFGs, it does not introduce new comparisons with the additional methods suggested by the reviewer.

Reviewer HG67 may increase their score to 6, as the authors provide additional baselines for the high-dimensional example and the proposed method continues to outperform these alternatives.

---

### Decision · Program_Chairs · 2026-01-26

Accept (Poster)